# Clustering of Tau fibrils impairs the synaptic composition of α3-Na+/K+-ATPase and AMPA receptors

Amulya Nidhi Shrivastava[1,2] (iD), Virginie Redeker[1] (iD), Laura Pieri[1], Luc Bousset[1], Marianne Renner[3], Karine Madiona[1], Caroline Mailhes-Hamon[2], Audrey Coens[1], Luc Buée[4], Philippe Hantraye[1], Antoine Triller[2,*] (iD) & Ronald Melki[1,**] (iD)

## Abstract

Tau assemblies have prion-like properties: they propagate from one neuron to another and amplify by seeding the aggregation of endogenous Tau. Although key in prion-like propagation, the binding of exogenous Tau assemblies to the plasma membrane of naïve neurons is not understood. We report that fibrillar Tau forms clusters at the plasma membrane following lateral diffusion. We found that the fibrils interact with the Na+/K+-ATPase (NKA) and AMPA receptors. The consequence of the clustering is a reduction in the amount of α3-NKA and an increase in the amount of GluA2-AMPA receptor at synapses. Furthermore, fibrillar Tau destabilizes functional NKA complexes. Tau and α-synuclein aggregates often co-exist in patients' brains. We now show evidences for cross-talk between these pathogenic aggregates with α-synuclein fibrils dramatically enhancing fibrillar Tau clustering and synaptic localization. Our results suggest that fibrillar α-synuclein and Tau cross-talk at the plasma membrane imbalance neuronal homeostasis.

**Keywords** cross-talk of pathogenic proteins; misfolding disease; protein aggregation and clustering; single-particle tracking; tauopathies
**Subject Categories** Neuroscience
**The EMBO Journal (2019) 38: e99871**

## Introduction

The six different isoforms of the microtubule-associated protein Tau (MAPT) neurons express have various functions, the main ones being in microtubules stabilization, cell morphogenesis, and axonal transport (Kovacs, 2015). These isoforms are the products of the alternative splicing of the pre-mRNA encoded by the 16 exons of *MAPT* gene. The six isoforms differ from each other by the presence or absence of one or two inserts (26 or 58 amino acid residues) in the N-terminal part of the protein and by the presence of either three (3R) or four (4R) repeated sequences (31–32 amino acid residues long) microtubule-binding motifs in the protein C-terminal part (Goedert *et al*, 2017). The phosphorylation of Tau is intimately linked to its assembly into high molecular weight oligomeric species (Buée & Delacourte, 1999). The latter, together with characteristic phosphorylation patterns, are hallmarks of several tauopathies including Alzheimer's disease (AD), Pick's disease, Tangle-only dementia, progressive supranuclear palsy, frontotemporal dementia, corticobasal neurodegeneration, argyrophilic grain disease, and frontotemporal dementia with parkinsonism linked to chromosome 17 (Kovacs, 2015; Goedert *et al*, 2017).

Experiments performed *in vitro* and using animal models support the notion that high molecular weight assemblies of Tau have prion-like properties. They are released by projections of affected neuronal cells, are taken up by unaffected cells, and are amplified by seeding the aggregation of endogenous Tau (Clavaguera *et al*, 2009; Brundin *et al*, 2010; Holmes *et al*, 2013; Pooler *et al*, 2013; Sanders *et al*, 2014; Iba *et al*, 2015; Takeda *et al*, 2015; Guo *et al*, 2016; Kaufman *et al*, 2016; Wegmann *et al*, 2016; Woerman *et al*, 2016; Narasimhan *et al*, 2017). Although key in the prion-like propagation of Tau assemblies, the binding to and molecular interactions of exogenous Tau assemblies with the plasma membrane of neurons are not fully characterized.

In this study, we document the interaction of exogenous fibrillar Tau (Fib-Tau) assemblies with neuron plasma membrane. We report that Fib-Tau clusters at the plasma membrane in a concentration- and time-dependent manner following lateral diffusion. We show that Fib-Tau resides within the clusters briefly. We identify 29 neuronal membrane proteins with extracellular domains that interact with Fib-Tau 3R and 4R assemblies. Of particular interest were

1 CEA, Institut François Jacob (MIRcen) and CNRS, Laboratory of Neurodegenerative Diseases (UMR9199), Fontenay-aux-Roses, France
2 Institut de Biologie de l'ENS (IBENS), École Normale Supérieure, INSERM, CNRS, PSL Research University, Paris, France
3 INSERM, UMR – S 839 Institut du Fer à Moulin (IFM), Sorbonne Université, Paris, France
4 CHU Lille, INSERM UMR-S 1172, JPArc, "Alzheimer & Tauopathies", Universite Lille, Lille, France
*Corresponding author. Tel: +33 1 44 32 35 47; E-mail: triller@biologie.ens.fr
**Corresponding author. Tel: +33 1 46 54 93 78; E-mail: ronald.melki@cnrs.fr

Na$^+$/K$^+$-ATPase (NKA) complex α1, α3 and β1 subunits, AMPA receptor GluA1 and GluA2 subunits, and NMDA receptor GluN1 and GluN2B subunits. We show that Fib-Tau clustering alters synaptic protein composition at excitatory synapses with a reduction in the amount of α3-NKA and an increase in the amount of AMPA receptor GluA2 subunit. The NMDA receptor subunit distribution was unaffected by Fib-Tau clustering. Single molecule trajectories of α3-NKA revealed displacement of this subunit from functional NKA pump. Our data suggest that Fib-Tau clusters destabilize functional NKA complexes, thus, reducing neuron's capacity to control membrane depolarization (Azarias *et al*, 2013). Therefore, we postulate that neuronal function is compromised by Fib-Tau clusters effect on NKA α3 subunit half-life and turnover. We further postulate that α3-NKA plays a role in Tau fibrils endocytosis and possibly in their subsequent amplification within neuron cytosol. This, together with GluA2-AMPA receptor redistribution and trapping within Fib-Tau clusters at excitatory synapses, may initiate deleterious signaling cascades.

Aβ, α-Syn, and Tau deposits are often observed in the brain of patients at late stages of PD and AD (Eisenberg & Jucker, 2012; Jellinger, 2012; Irwin *et al*, 2013; Morales *et al*, 2013; Moussaud *et al*, 2014). Here, we show that pathogenic α-Syn dramatically enhanced Fib-Tau clustering on neuronal plasma membrane. Ultimately, our results suggest that fibrillar α-Syn and Tau cross-talk at neuron plasma membrane may contribute to Alzheimer, Parkinson's, and related diseases onset or progression.

# Results

### Rapid uptake and redistribution of preformed Tau fibrils following injection in the CA1 region of the hippocampus

The injection of Tau assemblies into rodents' hippocampus was shown to trigger widespread deposition of pathological phosphory-lated Tau within weeks (Sanders *et al*, 2014; Takeda *et al*, 2015; Guo *et al*, 2016; Kaufman *et al*, 2016; Narasimhan *et al*, 2017). We first assessed the fate over 24 h of exogenous Tau fibrils. Preformed Tau-1N3R fibrils (hereafter referred to as Fib-Tau) were labeled with ATTO-550 dye and fragmented (see Materials and Methods and Appendix Figs S1, and S2A and B). Fib-Tau (3.9 μg in 1 μl) was injected in the right CA1-hippocampal cell body layer (Fig 1A, with permission from The Mouse Brain Library) of 1-month-old adult mice (C57BL6J strain; Rosen *et al*, 2003). Mice were sacrificed 8 or 24 h after injection. Low magnification (10×) fluorescence imaging revealed the presence of Fib-Tau (red) in the right corpus callosum region that is located between the hippocampus CA1 and cortex (Fig 1B, left) for all animals. The corpus callosum is rich in oligo-dendrocytes and myelinated axons facilitating communication between the two hemispheres of the brain (Meyer *et al*, 2018). High magnification (63×) images indicate that exogenous Fib-Tau binds along the processes (Fig 1B, right). Some accumulation within cells, possibly oligodendrocytes, of the corpus callosum could also be seen. No exogenous Fib-Tau could be detected in the corpus callosum of the left hemisphere of the brain within the time frame we explored. Furthermore, no differences in Fib-Tau distribution could be observed within 24 h in the right corpus callosum of the injected mice.

We assessed Fib-Tau distribution within regions adjacent to the injection site. Exogenous fibrillar Tau exhibited a diffused and clustered distribution in the stratum oriens and a pre-dominantly clustered distribution in the CA1 cell body layer 8 h after injection (Fig 1C and E, Appendix Fig S3). A weaker staining was observed 24 h after injection (Fig 1D and F, note the images presented in panel C and D were independently acquired images under non-identical exposure setting for display purpose) suggesting diffusion and potential clearance from stratum oriens. Background equivalent signal was detected in the cortex, stratum radiatum, and entorhinal cortex at all time points (Appendix Fig S3). To assess these changes, the density (number per μm$^2$) of exogenous Fib-Tau clusters (Fig 1G, obtained after thresholding, see Materials and Methods section) and total Fib-Tau fluorescence (Fig 1H, without thresholding) was quantified. Our measurements show a decrease in Fib-Tau clusters and total fluorescence over 24 h suggesting diffusion and/or the clearance of exogenous Tau fibrils.

### Time-, concentration-, and lateral diffusion-dependent clustering of Fib-Tau on neuronal plasma membrane

Exogenous fibrillar Tau binding and clustering *in vivo* (Fig 1) are reminiscent of that of amyloid-β (Aβ) oligomers (Renner *et al*, 2010) and fibrillar α-Syn (Shrivastava *et al*, 2015). To assess Fib-Tau clustering in details, we used mouse primary neuronal cultures. Unless otherwise stated, the experiments were performed on densely mature (DIV 21–24) primary hippocampal neuronal cultures with well-developed synapses and network activity (Ivenshitz & Segal, 2010). Neurons were exposed for 1 h to exogenous Fib-Tau (red, 0.36 nM expressed in particle concentration—see Appendix Fig S1) and labeled with anti-MAP2 (dendrites, blue) and anti-endogenous Tau (axons, green) antibodies (Appendix Fig S4A). Our observations reveal the presence of clusters of exogenous Fib-Tau, with some background diffused labeling, both on axons and on dendrites (Appendix Fig S4A). This indicates that there is no preferential binding to either dendrites or axons. Similar clustering was observed on HEK cell membranes upon exposure (10 min) to Fib-Tau (Appendix Fig S4B).

We next assessed the clustering of exogenous ATTO-488-labeled Fib-Tau (Appendix Fig S2D), in time-dependent (0.5 s to 4 h at 0.36 nM concentration, a concentration where low-density labeling of Tau was achieved and neurons could be identified) and concentration-dependent (0.36 nM to 1,000 nM for 1-h exposure) manners. Representative images are shown in Fig 2A, and quantification is displayed in Fig 2B and C. Within the concentration range 0.36–0.72 nM, only the density (number per μm$^2$) of exogenous Fib-Tau clusters increased with time (Fig 2A, top panel, Fig 2B). The size of the clusters (measured by fluorescence intensity) did not increase in a significant manner (Fig 2C). At higher concentrations (1.8 nM and higher), both the density and size of Fib-Tau clusters increased with time.

The clustering of exogenous Fib-Tau was next assessed by single-particle tracking using quantum dots (SPT-QD) approach. Neurons were exposed (0.36 nM) to biotin-labeled Fib-Tau (0.36 nM, Appendix Fig S2E) for 10 or 60 min followed by labeling a small fraction of the exogenous biotin-labeled fibrils with strepta-vidin-QD-655. Representative single molecule trajectories (red) of

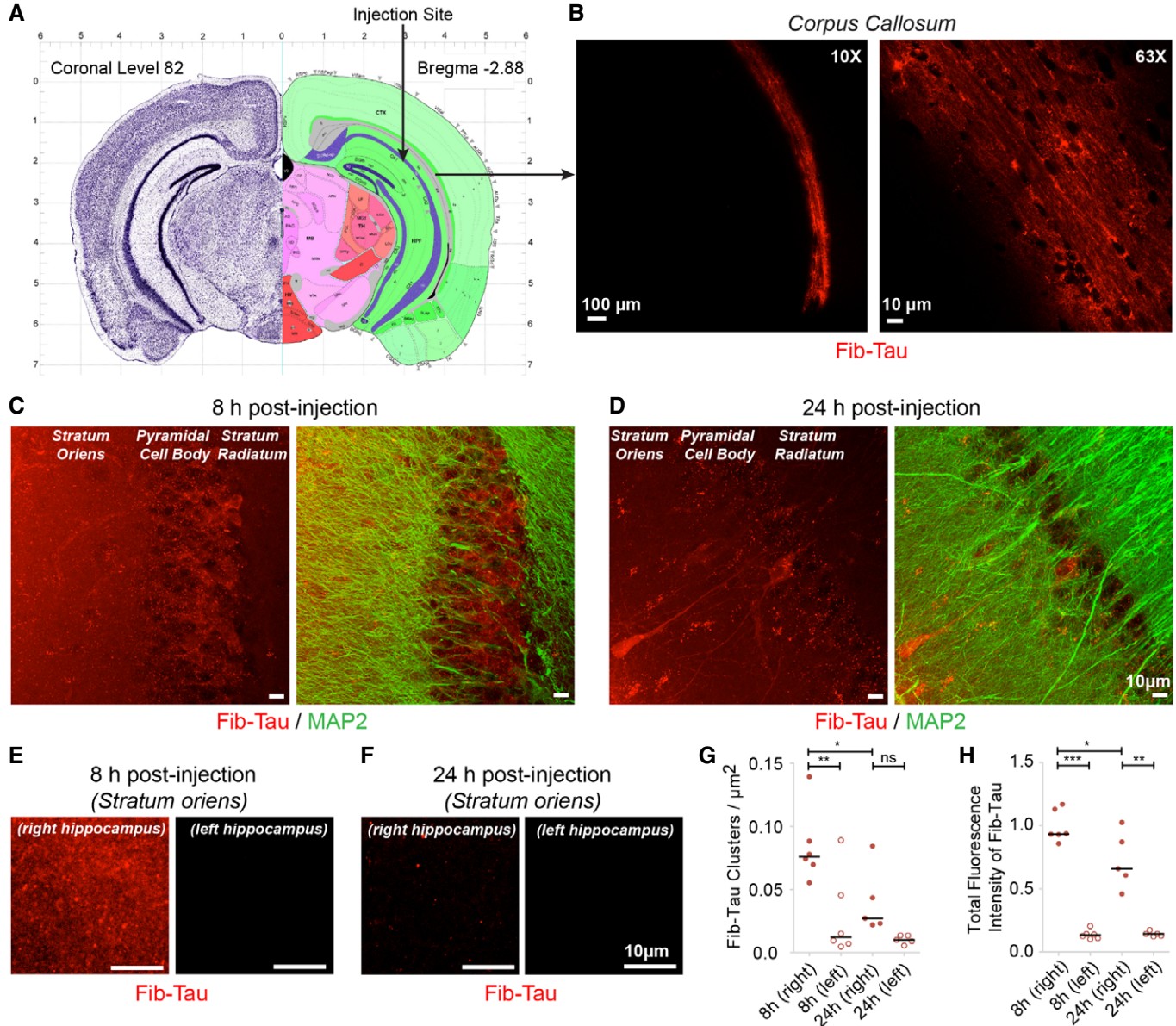

**Figure 1. Rapid uptake and redistribution of preformed Fib-Tau following injection in the CA1 region of the hippocampus.**

A, B Coronal brain section, bregma: −2.88 from The Mouse Brain Atlas shows the site of injection in the CA1 cell body layer of the right hippocampus (A). Eight hours post-injection, majority of the injected Fib-Tau-1N3R-ATTO-550 (red) is taken up by the corpus callosum of the injected side (B). Images are shown at low (left, 10×) and high (right, 63×) magnification.

C–H Representative images of the distribution of Fib-Tau (red) 8 h (C, E) and 24 h (D, F) after injection. Dendrites are immuno-labeled using MAP2 antibody (green). Eight hours after injection, both diffused and clustered distribution of exogenous Fib-Tau are observed in stratum oriens and CA1 cell body region (C, E, Appendix Fig S3). No exogenous Fib-Tau is detected in the adjoining cortex, stratum radiatum, and entorhinal cortex (see Appendix Fig S3). Twenty-four hours after injection, clustered distribution is primarily restricted to the pyramidal (CA) cell body layer (D). Uptake of Fib-Tau is seen in some cells in both stratum oriens and pyramidal cell body layer (D). The number of Fib-Tau clusters per $\mu m^2$ within stratum oriens decreases between 8 and 24 h post-injection (G). A similar decrease in the total fluorescence (clusters + diffused) is observed indicating the diffusion and/or clearance of exogenous Fib-Tau (H). Paired *t*-test performed between "8/ 24 h-right" and "8/24 h-left"; unpaired *t*-test performed between "8 h-left" and "24 h-left" columns (*n* is number of animals: 6 for 8 h and 5 for 24 h); ***$P < 0.001$, **$P < 0.01$, *$P < 0.05$, ns = not significant. Note: The images presented in panels (C and D) were independently acquired images under non-identical exposure setting at low magnification for display purpose; panels (E and F) are shown for same exposure and are used for quantification.

Data information: Scale bar, 100 μm in the left panel in (B), 10 μm everywhere else.

biotin-labeled Fib-Tau (Fig 2D, red) show they explore large region area after 10 min, while their diffusion is confined after 60 min. This global slowdown in the diffusion coefficient (Fig 2E) and decrease in explored area (Fig 2F, indicative of mean squared displacement, MSD) suggest that Fib-Tau experience molecular interactions leading to the formation of clusters on the plasma

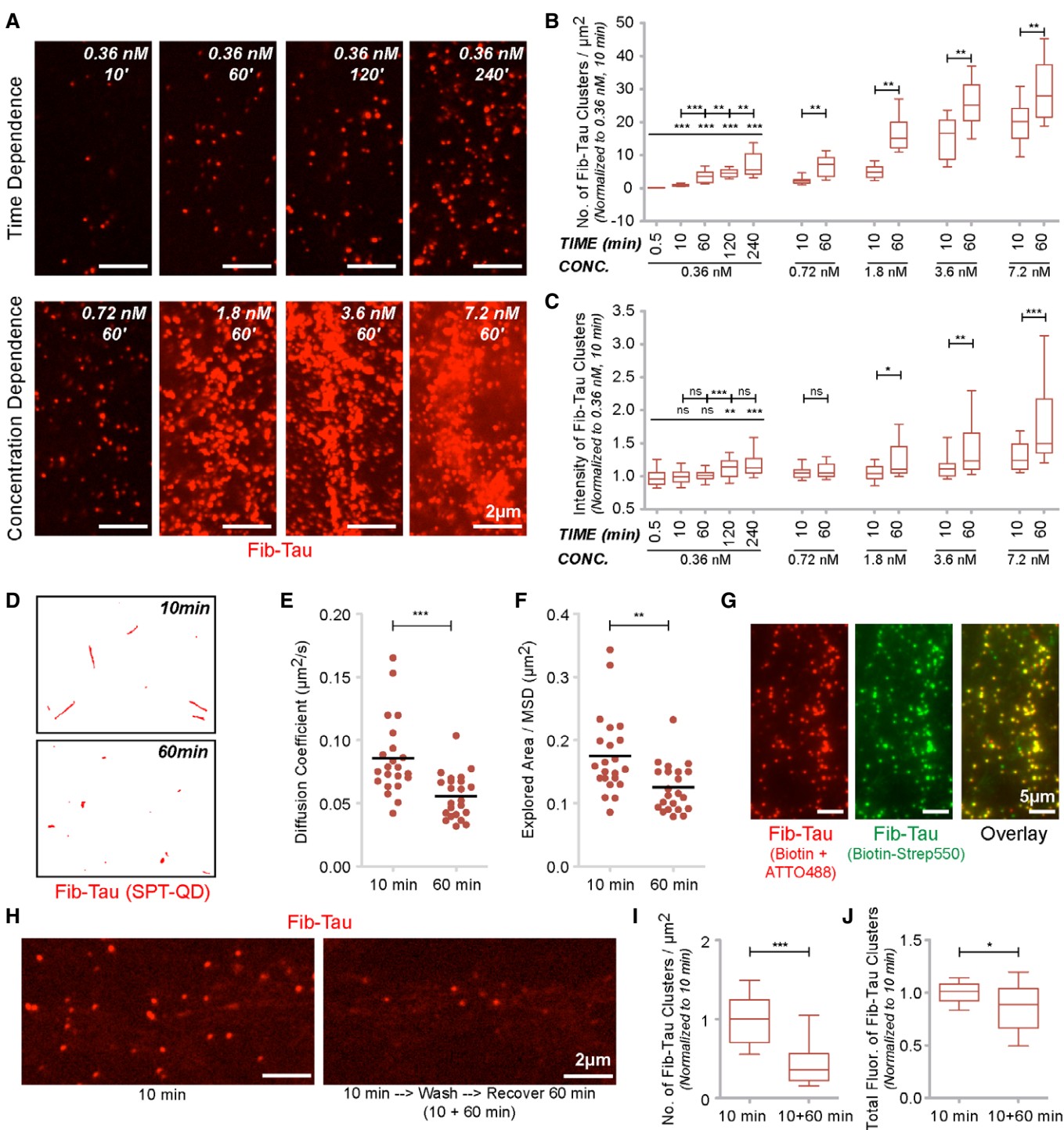

**Figure 2.**

membrane. To confirm that Fib-Tau clusters are located on the outer side of the plasma membrane, neurons were exposed (60 and 240 min) to preform Fib-Tau labeled simultaneously with both biotin and ATTO-488 (0.36 nM, Fig 2G, Appendix Fig S5, red). Labeling of the biotin with streptavidin-550 in live neurons (Fig 2G, green) shows that most, if not all, Fib-Tau clusters are at the cell surface (overlay, Fig 2G, Appendix Fig S5).

*In vivo* experiments (Fig 1E–H) suggest a substantial diffusion/ clearance of exogenous Fib-Tau 24 h after injection within the hippocampus. To determine whether this also occurs *in vitro*, primary neurons were exposed (0.36 nM) for 10 min to ATTO-550-labeled Fib-Tau, the excess of fibrils was washed away, and the cells were imaged immediately or allowed to recover for 60 min. A significant number of Fib-Tau clusters were observed

◀

**Figure 2.  Time-, concentration-, and lateral diffusion-dependent clustering of Fib-Tau on neuronal plasma membrane.**

A–C   Time- and concentration-dependent clustering of Fib-Tau in primary neurons. Representative images are shown for certain conditions to illustrate Fib-Tau clustering time dependence (A, top row) and concentration dependence (A, bottom row). Quantification of the number of Fib-Tau clusters per $\mu m^2$ (B) or fluorescence intensity of clusters (C, indicating size, refer to Materials and Methods). At low concentrations (up to 0.72 nM), the density of clusters increased with time (between 10 and 60 min) but the increase in intensity was small. At high concentrations of Fib-Tau ($\geq$ 1.8 nM), both density and size increased with increasing time. Box-plot represents median, interquartile range, and 10–90% distribution; one-way ANOVA with Dunnett's *post hoc* test, number of images analyzed from three cultures (from left to right: 25, 25, 25, 25, 70, 45, 45, 45, 45, 45, 30, 40, 40, 40, 40, and 40 images).

D–F   Single-particle tracking using quantum dots (SPT-QD) of biotin-tagged Fib-Tau. Representative single molecule trajectories of Fib-Tau following 10- or 60-min exposure are shown (D). Note after 60-min exposure (0.36 nM), single molecules are more confined suggesting they are trapped and clustered. Quantification of diffusion coefficient (E) and explored area (F, extracted from mean squared displacement (MSD), see Materials and Methods) shows that both these parameters decrease after 60-min exposure to Fib-Tau. Unpaired *t*-test, *n* is averaged value per cells imaged in three experiments (10 min: 22, 60 min: 23).

G   Neurons were exposed for 60 min to Fib-Tau (0.36 nM) labeled with both biotin and ATTO-488 (red). Cell surface-exposed biotin was labeled using streptavidin-550 (green) followed by live imaging. Note that most of the clusters of ATTO-488 (red) are co-labeled with streptavidin-550 (green) indicating that the clusters are at the cell surface.

H–J   Clearance of Tau clusters from neurons. Neurons were exposed (0.36 nM) to ATTO-550-labeled Fib-Tau for 10 min, and the unbound fibrils were washed. Cells were fixed immediately (10 min) or allowed to recover in culture medium for 60 min. Two representative images (H) and quantifications (I, J) show that following 60-min recovery most of the Tau clusters disappear/dissociate as indicated by a decrease in their density. Box-plot represents median, interquartile range, and 10–90% distribution; unpaired *t*-test, *n* is number of images analyzed from three cultures (49 images).

Data information: *$P < 0.05$; **$P < 0.01$; ***$P < 0.001$; ns = not significant. Scale bar, 5 $\mu m$ in (G), 2 $\mu m$ everywhere else.

on the surface of neurons upon imaging immediately (10 min) after removing the excess of exogenous fibrils (Fig 2H, left). In contrast, few clusters were detected when the cells were allowed to recover for 60 min (Fig 2H, right). A quantitative assessment of Fib-Tau cluster density and fluorescence confirms these observations (Fig 2I and J). This suggests either that Fib-Tau clusters dissociate from neuronal membranes over time or that they are taken up by the cells and processed. We conclude from these observations that (i) Fib-Tau clusters in a time- and concentration-dependent manner at the surface of neurons, (ii) clustering slows down the diffusion of Fib-Tau, and (iii) Fib-Tau either dissociates from neuronal membrane with time or are taken up and processed within the cells.

**Dynamic assessment of the clustering of exogenous Fib-Tau on neuron plasma membranes**

The diffusion-dependent clustering of exogenous Fib-Tau at the surface of neurons exhibits similarities with that of Aβ oligomers and α-Syn assemblies with two exceptions. The time-dependent increase in cluster size for Fib-Tau (Fig 2B) is much slower than that previously measured for Aβ oligomers (Renner *et al*, 2010) and α-Syn assemblies (Shrivastava *et al*, 2015), and the dissociation and/or clearance of Fib-Tau, both *in vitro* and *in vivo,* is much faster than that observed for the other two assemblies (Figs 1E and F, and 2H). These data suggest that the interaction of Fib-Tau with the plasma membrane is highly dynamic and the clusters the fibrils form are unstable. To assess in a quantitative manner the amounts of clustered and non-clustered Fib-Tau, we performed super-resolution STORM (stochastic optical reconstruction microscopy) analysis on neurons exposed to ATTO-647-labeled Fib-Tau (0.36 nM, Appendix Fig S2F) for increasing time (10, 60, 120, and 240 min). Rendered images obtained with pixel size of 5 nm revealed a large proportion of non-clustered ATTO-647-labeled Fib-Tau at all time points (Fig 3A). Fib-Tau distribution was non-uniform over the entire surface of neurons. The total number of "single-particle detection events" per $\mu m^2$ of neuronal surface increased with time (Fig 3B). The total number of clusters identified using DBSCAN (density-based spatial clustering of applications with noise; Ester

*et al*, 1996; Malkusch *et al*, 2012) approach increased with time (Fig 3C) in agreement with the data presented in Fig 2B. The radius of Fib-Tau clusters (Fig 3D) and the number of events per cluster (Fig 3E), however, were unchanged, in agreement with the observation presented in Fig 2C. We also assessed the proportion of single-particle detection events within Fib-Tau clusters as a function of time. The data (Fig 3F) show that 50% of Fib-Tau reside within the clusters at any given time point and concentration of membrane-bound Fib-Tau. This contrasts with the behavior of fibrillar α-Syn, where about 90% of single fibrils were within clusters (Shrivastava *et al*, 2015).

**Identification of membrane proteins that interact with extracellularly applied Fib-Tau assemblies**

A proteomic screening was performed to identify membrane proteins interacting with extracellularly applied Fib-Tau. Preformed biotin-labeled Fib-Tau (Appendix Fig S2E) was used as described in the Materials and Methods section to identify specific neuronal membrane protein partners as illustrated (Fig 4A). Pure cultures of cortical neurons were exposed to biotin-Fib-Tau (14.4 nM, to achieve rapid binding and saturate the binding sites) for 10 min. Biotin-Fib-Tau and associated proteins were pulled down from whole-cell lysates using streptavidin immobilized on magnetic beads, eluted with Laemmli buffer and in-gel trypsin digested. The resulting peptides were identified by nanoLC-MS/MS (LC: liquid chromatography; MS/MS: tandem mass spectrometry). Control samples were prepared from neurons unexposed to Fib-Tau. Protein abundance was assessed by a label-free quantitative proteomic method using spectral counting. We identified 103 and 957 neuronal protein partners from cells used as control or exposed to exogenous Fib-Tau, respectively (Fig 4B). Several intracellular proteins were identified possibly following their interaction with endocytosed Fib-Tau (Flavin *et al*, 2017) and/or interaction following cell disruption during protein extraction. A total of 372 synaptic and membrane proteins were identified (Fig 4C, Appendix Table S1). Twenty-nine proteins had extracellular domains. The rest of the proteins we identified are scaffolding and signaling proteins. Indeed, pull-down experiments identify not only the direct partners of Fib-Tau but also

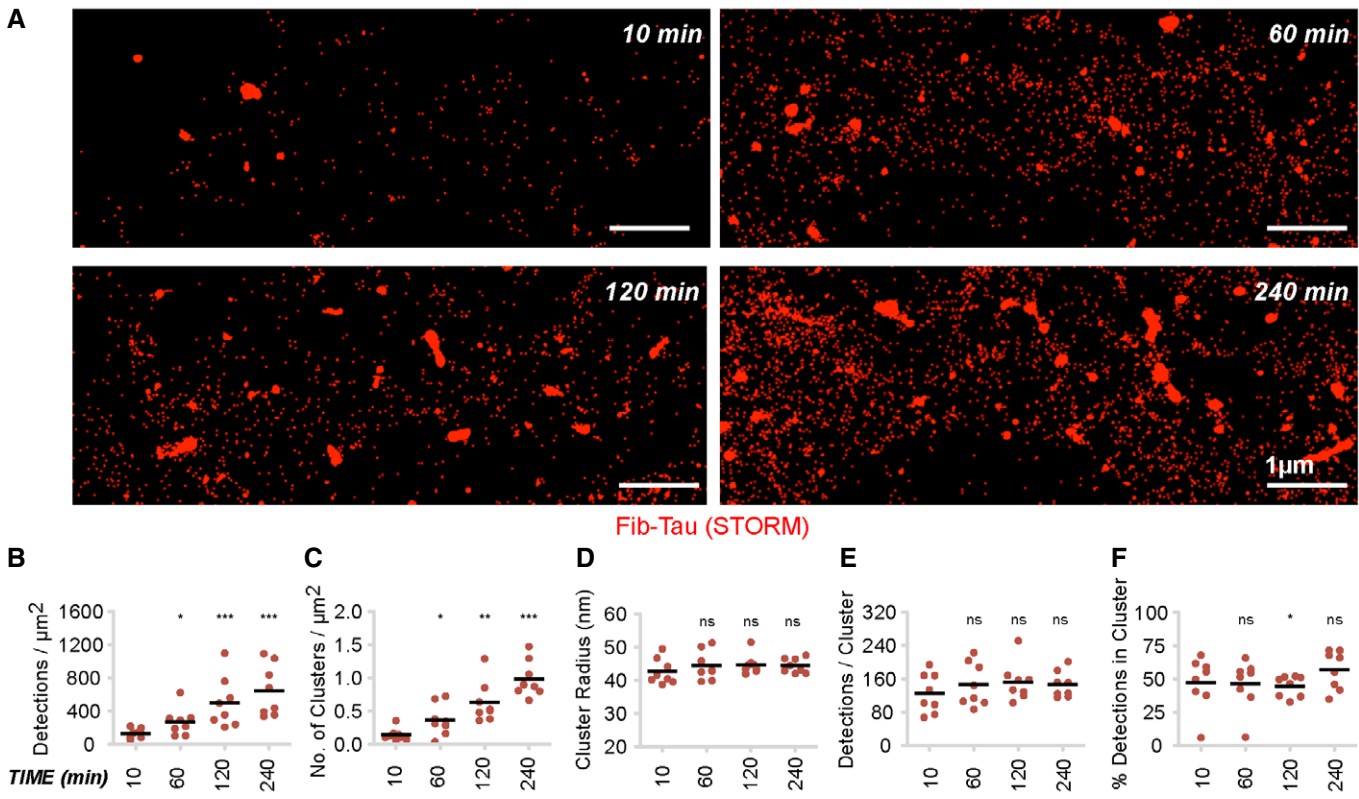

**Figure 3. Equilibrium between clustered and free Fib-Tau on neuronal plasma membrane.**

A    Super-resolution STORM images of neuronal membrane after exposure to ATTO-647N-labeled Fib-Tau (0.36 nM). Primary neurons were fixed 10, 60, 120, and 240 min after exposure to ATTO-647N-labeled Fib-Tau. Images rendered with a pixel size of 5 nm show both clustered and diffused (non-clustered) distribution of exogenous Fib-Tau molecules. Scale bar, 1 μm.

B–F    Quantification of various parameters from the super-resolution images. Averaged values per image are plotted. Increased "total detections per μm²" (B) and "cluster density" (C) of Fib-Tau bound to neurons with increasing exposure time is observed. No change in the "size of cluster" (D) or "number of molecules within a cluster" (E) is seen with increased exposure time. The proportion of single molecule events detected within clusters remained nearly constant (~50%) in all conditions (F) indicating an equilibrium between clustered and non-clustered Fib-Tau single molecules. Unpaired *t*-test to compare difference with 10 min and *n* is number of images (eight images). \*$P < 0.05$; \*\*$P < 0.01$; \*\*\*$P < 0.001$; ns = not significant.

the protein complexes that interact with Fib-Tau binders. The list of membrane proteins with extracellular domains that interact with Fib-Tau is given in Fig 4D.

Among the proteins with extracellular domains, we identified are key neurotransmitter receptors namely GluA1 and GluA2 subunits of AMPA receptors (identified through 8 and 12 unique peptides, respectively, Appendix Fig S6), and GluN1 and GluN2B subunits of NMDA receptors (identified through 16 and 12 unique peptides, respectively, Appendix Fig S6). This finding strongly suggests that exogenous Fib-Tau may directly interact with excitatory PSD (post-synaptic density) super-complex (Loh *et al*, 2016). Fib-Tau also interacts with NKA (sodium–potassium ATPase) α3 subunit (identified through 27 peptides, 15 unique, Appendix Fig S6), in a manner similar to fibrillar α-Syn (Shrivastava *et al*, 2015) and oligomeric Aβ (Ohnishi *et al*, 2015). Fib-Tau also interacts with the α1 subunit (identified through 17 peptides, 7 unique) and the β1 subunit (identified through 6 unique peptides) of NKA. The interaction between GluA1 and GluA2 subunits of AMPA receptor, GluN1 and GluN2B subunits of NMDA receptor, and the α1, α3, and β1 subunits of NKA was confirmed using chemical cross-linking combined to nanoLC-MS/MS (Fig 4D, last column). For this, membrane proteins were

cross-linked using the cleavable cross-linker DTSSP (3,3′-dithiobis (sulfosuccinimidylpropionate)) in neurons exposed for 10 min to Fib-Tau-1N3R (14.4 nM). Proteins interacting with Fib-Tau were pulled down, reduced, alkylated, cleaved with trypsin, and subjected to MS as described above.

The interaction of Fib-Tau with NKA, AMPA, and NMDA receptors was further assessed using co-immunoprecipitation experiments (Fig 4E). When α3-NKA, GluA2-AMPA, or GluN1-NMDA receptors were immunoprecipitated using specific antibodies after exposure of neurons to Fib-Tau (Fig 4E), Tau could be co-immuno-precipitated (Fig 4E). The Western and slot blot data show that α3-NKA and GluA2-AMPA interact strongly with Fib-Tau, while GluN1-NMDA receptors interact weakly (Fig 4E). This experiment mirrors the biotinylated fibrillar Tau pull-down experiments (Fig 4A–D) and demonstrates interaction of Fib-Tau and α3-NKA and the GluA2-AMPA receptors.

The interactomes of the membrane proteins with extracellularly exposed polypeptides (in yellow) pulled down with Fib-Tau with the post-synaptic proteins (in cyan) derived from the String database (String v10, https://string-db.org/; Szklarczyk *et al*, 2015) is presented in Fig 4F.

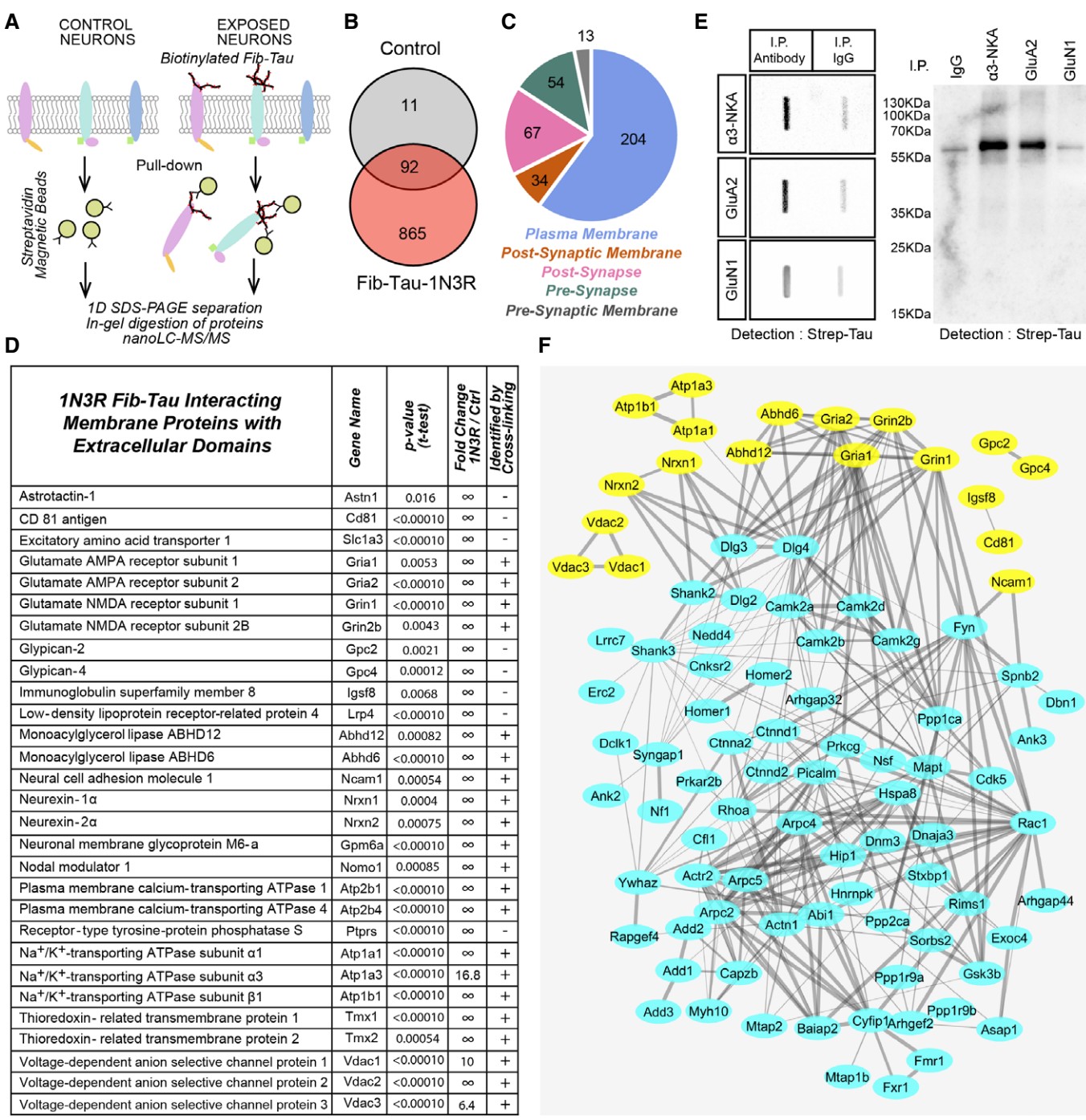

**Figure 4.**

| 1N3R Fib-Tau Interacting Membrane Proteins with Extracellular Domains | Gene Name | p-value (t-test) | Fold Change 1N3R / Ctrl | Identified by Cross-linking |
|---|---|---|---|---|
| Astrotactin-1 | Astn1 | 0.016 | ∞ | - |
| CD 81 antigen | Cd81 | <0.00010 | ∞ | - |
| Excitatory amino acid transporter 1 | Slc1a3 | <0.00010 | ∞ | - |
| Glutamate AMPA receptor subunit 1 | Gria1 | 0.0053 | ∞ | + |
| Glutamate AMPA receptor subunit 2 | Gria2 | <0.00010 | ∞ | + |
| Glutamate NMDA receptor subunit 1 | Grin1 | <0.00010 | ∞ | + |
| Glutamate NMDA receptor subunit 2B | Grin2b | 0.0043 | ∞ | + |
| Glypican-2 | Gpc2 | 0.0021 | ∞ | - |
| Glypican-4 | Gpc4 | 0.00012 | ∞ | - |
| Immunoglobulin superfamily member 8 | Igsf8 | 0.0068 | ∞ | - |
| Low-density lipoprotein receptor-related protein 4 | Lrp4 | <0.00010 | ∞ | - |
| Monoacylglycerol lipase ABHD12 | Abhd12 | 0.00082 | ∞ | + |
| Monoacylglycerol lipase ABHD6 | Abhd6 | <0.00010 | ∞ | + |
| Neural cell adhesion molecule 1 | Ncam1 | 0.00054 | ∞ | + |
| Neurexin-1α | Nrxn1 | 0.0004 | ∞ | + |
| Neurexin-2α | Nrxn2 | 0.00075 | ∞ | + |
| Neuronal membrane glycoprotein M6-a | Gpm6a | <0.00010 | ∞ | + |
| Nodal modulator 1 | Nomo1 | 0.00085 | ∞ | + |
| Plasma membrane calcium-transporting ATPase 1 | Atp2b1 | <0.00010 | ∞ | + |
| Plasma membrane calcium-transporting ATPase 4 | Atp2b4 | <0.00010 | ∞ | + |
| Receptor-type tyrosine-protein phosphatase S | Ptprs | <0.00010 | ∞ | - |
| Na⁺/K⁺-transporting ATPase subunit α1 | Atp1a1 | <0.00010 | ∞ | + |
| Na⁺/K⁺-transporting ATPase subunit α3 | Atp1a3 | <0.00010 | 16.8 | + |
| Na⁺/K⁺-transporting ATPase subunit β1 | Atp1b1 | <0.00010 | ∞ | + |
| Thioredoxin-related transmembrane protein 1 | Tmx1 | <0.00010 | ∞ | + |
| Thioredoxin-related transmembrane protein 2 | Tmx2 | 0.00054 | ∞ | + |
| Voltage-dependent anion selective channel protein 1 | Vdac1 | <0.00010 | 10 | + |
| Voltage-dependent anion selective channel protein 2 | Vdac2 | <0.00010 | ∞ | + |
| Voltage-dependent anion selective channel protein 3 | Vdac3 | <0.00010 | 6.4 | + |

Finally, the membrane proteins interacting with extracellularly applied Fib-Tau-1N4R isoform were also identified following the strategy described above for Fib-Tau-1N3R isoform fibrils. We identified 90 and 1,063 neuronal protein partners from cells used as control or exposed to exogenous 1N4R Fib-Tau, respectively (Fig EV1A). A total of 379 synaptic and membrane proteins were identified (Fig EV1B, Appendix Table S2). The list of membrane proteins with extracellular domains given in Fig EV1C reveals that Fib-Tau-1N4R assemblies also bind to GluN1 and GluN2B

subunits of NMDA receptors and the α1, α3, and β1 subunits of NKA. The interaction of Fib-Tau-1N4R with AMPA receptors was weaker since only the GluA2 subunit was identified (Fig EV1C). This is supported by the observation that α3-NKA and GluN1-NMDA receptors but not GluA2-AMPA receptors co-immunoprecipitated with Fib-Tau-1N4R (Fig EV1D and E). Overall, 1N3R and 1N4R Fib-Tau synaptic and membrane protein partners are listed back to back (Appendix Table S3) to illustrate the high degree of overlap.

**Figure 4.   Identification of intrinsic neuronal membrane proteins interacting with extracellularly applied Fib-Tau-1N3R.**

A   Strategy used to purify and identify neuron intrinsic membrane proteins with extracellular domain that interact specifically with Fib-Tau-1N3R. Fib-Tau was labeled 1 h with 10 molar equivalents of NHS-S-S-Biotin. Mouse cortical neuron cultures were exposed for 10 min to biotinylated Fib-Tau (14.4 nM). Fresh protein extracts from those neurons were incubated with streptavidin magnetic beads to pull down Fib-Tau together with their specific protein partners. Unexposed neuron extracts were used as a control. Proteins bound to the streptavidin magnetic beads were eluted with Laemmli buffer and subjected to short migration on a SDS–PAGE gel. After Coomassie blue staining, proteins were subjected to in-gel digestion using trypsin and subsequently identified by nanoLC-MS/MS analysis, using a nanoLC-TripleTOF mass spectrometer. Relative quantification between control and exposed neuron samples was performed using a label-free MS-based approach. Six independent replicates were analyzed.

B   Venn diagram of 968 proteins identified in Fib-Tau pull-downs only (red), in control pull-downs only (gray), or in both samples (overlap). Of the 92 proteins identified in both samples, 45 proteins were significantly enriched in Fib-Tau pull-downs (*t*-test with *P*-values < 0.05, Benjamini–Hochberg, fold change > 2).

C   Distribution of the 372 synaptic and membrane protein interactors of Fib-Tau identified in the pull-down experiments. Locations of proteins at the levels of subcellular structures were annotated using the Gene Ontology Cell Component annotation tool of AMIGO 2 (http://amigo.geneontology.org/amigo/landing). Distribution of Fib-Tau interactors in the plasma membrane, pre-synaptic membrane, post-synaptic membrane, pre-synapse, and post-synapse is shown.

D   List of synaptic and plasma membrane proteins with extracellular domains significantly enriched in pull-downs from neurons exposed to Fib-Tau. For each identified protein, the name of the protein, the gene name, the *P*-value (*t*-test with Benjamini–Hochberg correction), and the fold change corresponding to the ratio of spectral counts between exposed neuron and control samples are given. In an independent analysis, after 10-min exposure of neurons to biotinylated Fib-Tau, a cross-linking step was performed during 20 min using 1 mM of DTSSP added in the culture medium, in order to cross-link the protein complexes formed at the cell surface using a membrane impermeable cross-linker. After cross-linking, proteins were analyzed and identified exactly as non-cross-linked samples. Proteins identified with at least two peptides are labeled "+", and the other are labeled "−".

E   Co-immunoprecipitation of exogenous biotin-labeled Fib-Tau with α3-NKA, GluA2, and GluN1. α3-NKA, GluA2, and GluN1 were immunoprecipitated using specific antibodies as described in the Materials and Methods section. The presence of Fib-Tau in the immunoprecipitate was assessed using a slot blot apparatus and nitrocellulose membranes probed with streptavidin-HRP. A 2.4-, 2.3-, and 1.8-fold enrichment in Tau band intensity is observed in α3-NKA, GluA2, and GluN1 immunoprecipitates, respectively, compared to controls performed with pre-immune goat or rabbit IgGs. Co-immunoprecipitates of exogenous biotin-labeled Fib-Tau with anti-α3-NKA-, GluA2-, and GluN1-specific antibodies were also subjected to SDS–PAGE and Western blot analysis. The presence of Fib-Tau in the immunoprecipitates was assessed by probing the nitrocellulose membranes with streptavidin-HRP. Fib-Tau co-immunoprecipitates with α3-NKA and GluA2-AMPA receptor but not with GluN1-NMDA receptor.

F   Network describing the interconnectivity of intrinsic membrane proteins extracellularly exposed (presented in panel D, labeled in yellow) and post-synaptic proteins (proteins with a post-synapse or a post-synaptic membrane annotation, presented in panel C and labeled in blue) that interact with Fib-Tau-1N3R. This Fib-Tau-1N3R interactome was input in the String database (String v10, https://string-db.org/) and exported to Cytoscape (version 3.5.1 at http://www.cytoscape.org/) to visualize interactions between the identified proteins. A total of 121 proteins were evaluated. We set parameters to only detect interactions that were validated experimentally or described in databases. The thickness of the line corresponds to the confidence of interaction (thin lines, > 0.4; medium lines, > 0.7; thick lines, > 0.9).

## Clustering of Fib-Tau at excitatory synapses redistributes α3-NKA and AMPA but not NMDA receptors

We next assessed Fib-Tau distribution at synapses. Neurons exposed to ATTO-550-labeled Fib-Tau (0.36 nM, 10 or 60 min) were immuno-stained with antibodies directed against homer (green, excitatory synapses) and gephyrin (blue, inhibitory synapses; Fig 5A). At this concentration and within this time frame, 10–20% of Fib-Tau clusters co-localized with excitatory synapses (Fig 5B). Almost no co-localization was observed with inhibitory synapses (Fig 5B; that accounted for about 10–15% of total synapses in mouse primary cultures at DIV 21–24). These observations along with proteomics data indicate that Fib-Tau clusters specifically at excitatory synapses in hippocampal neurons. To verify this, we quantified Fib-Tau binding at synapses in inhibitory synapse-enriched spinal cord neurons (Appendix Fig S7). No synapse-specific binding of Fib-Tau was observed in these cells indicating that Fib-Tau clustering at excitatory synapse is also cell-type dependent. Lastly, no differences in the clustering properties (e.g., density, Fig EV2A and B) or synaptic co-localization (Fig EV2C) were observed between Tau-1N3R and 1N4R fibrils.

We next performed quantitative co-localization studies to assess the consequences of the interaction of Fib-Tau-1N3R and 1N4R with GluA1 and GluA2 subunits of AMPA receptors, GluN1 and GluN2B subunits of NMDA receptors, and the α3-NKA at excitatory synapses. Neurons were exposed to ATTO-550-labeled Tau-1N3R or 1N4R fibrils (0.36 nM, 60 min), and the cells were immuno-labeled with anti-α3-NKA, GluA1-AMPA, GluA2-AMPA, GluN1-NMDA, GluN2B-NMDA, and homer or PSD95 antibodies (Fig 5C). Co-localization between Fib-Tau (red), homer/PSD95 (blue), and α3-NKA,

AMPA, and NMDA receptors (green) was observed within excitatory synapses (Fig 5C, arrows). The total fluorescence intensity of receptor subunits co-localizing with synapses (i.e., receptor clusters that co-localize with homer/PSD95 clusters) was quantified. A reduction in the amount of synaptic α3-NKA (−19.8%, difference between median values) and an increase in the amount of GluA2-AMPA (+12.9%) receptors at synapses following exposure to Fib-Tau were observed suggesting an alteration in the synaptic protein composition for these two proteins. No change in GluA1-AMPA (−1.1%) receptor and the GluN1-NMDA (−3.2%) and GluN2B-NMDA (2.6%) receptor was observed. Similar observations were made upon exposure of neurons to fibrillar Tau 1N4R and assessment of the redistribution of α3-NKA, GluA1, GluA2, GluN1, and GluN2B (Fig EV2D).

To determine to what extent fibrillar Tau co-localizes with the synaptic partners we identified *in vivo*, Fib-Tau was injected within the hippocampus and their co-localization with α3-NKA, GluA1, GluA2, GluN1, GluN2B was assessed by immuno-labeling brain sections. Homer staining revealed that 10–25% Fib-Tau clusters co-localized or were found apposed (within 150 nm distance) to excitatory synapses (Appendix Fig S8). The figure for α3-NKA was 10–30%. Some co-localization between Fib-Tau clusters and the glutamate receptor AMPA and to a lesser extent NMDA was also observed (Appendix Fig S8).

## Fib-Tau diffusion traps GluA2-AMPA receptors and increases action-potential-induced Ca²⁺ response

The accumulation of the GluA2 subunit of AMPA receptors within excitatory synapses (Fig 5C and D), the observation that GluA2 is

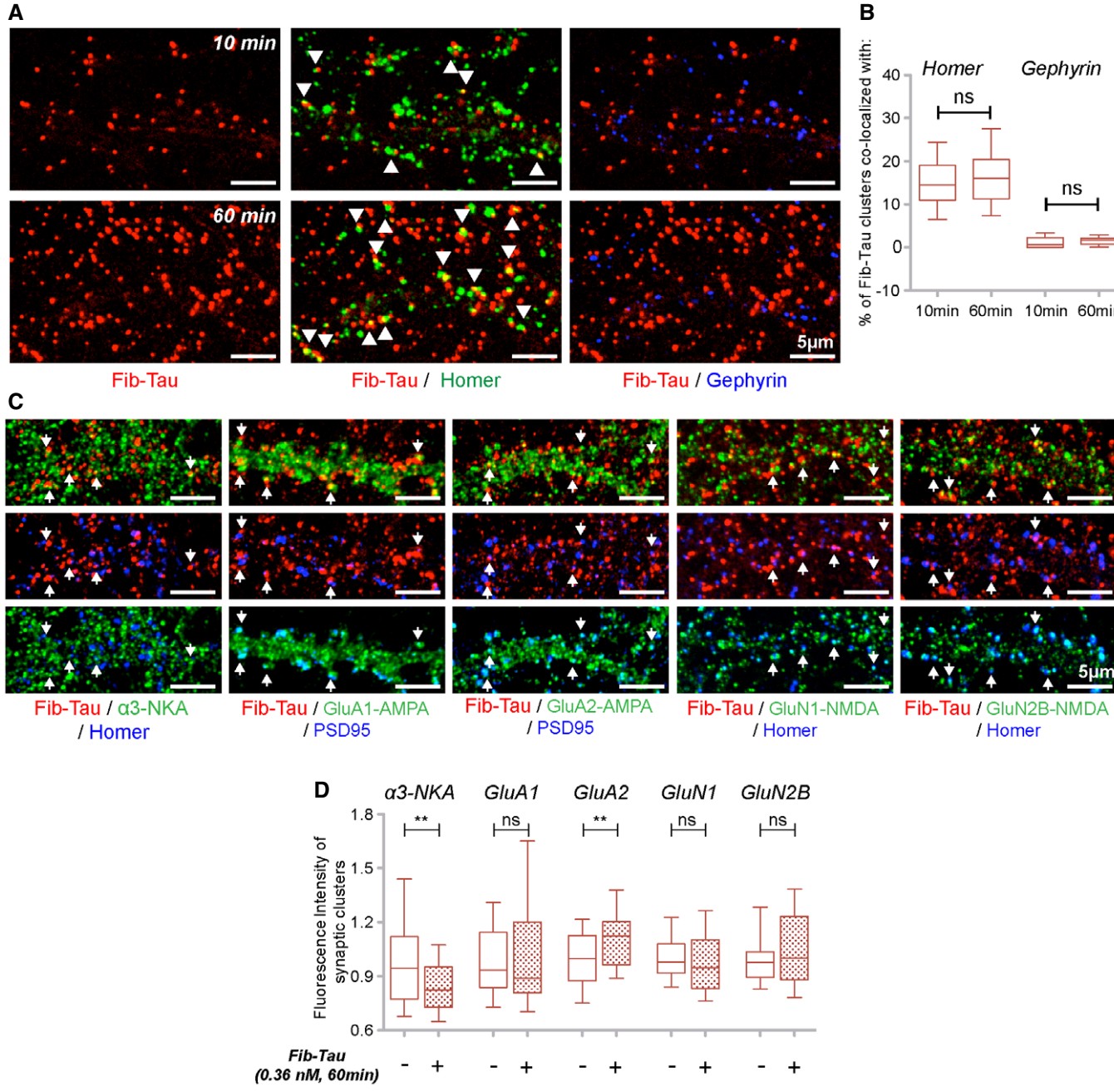

**Figure 5. Clustering of Fib-Tau at excitatory synapses redistributes α3-NKA and AMPA but not NMDA receptors.**

A, B    Primary neurons were exposed to Fib-Tau-ATTO-550 (0.36 nM, red) for 10 (top row) or 60 min (bottom row), and excitatory (homer, green) and inhibitory synapses (Gephyrin, blue) were immuno-labeled. Fib-Tau clusters co-localized and/or apposed to homer (arrowheads) but not Gephyrin (A). Quantitative analysis shows that 10–20% of Fib-Tau clusters are localized at homer containing synapses (B). Box-plot represents median, interquartile range, and 10–90% distribution, unpaired *t*-test to compare difference between 10 and 60 min, and *n* is number of images analyzed from three experiments (10 min: 49; 60 min: 50). Scale bar, 5 μm.

C, D    Exposure of neurons to Fib-Tau-ATTO-550 (0.36 nM, 60 min, red) and immuno-labeling of excitatory synapse (anti-rabbit-homer or anti-mouse-PSD, blue) and α3-NKA or GluA1-AMPA or GluA2-AMPA or GluN1-NMDA or GluN2B-NMDA subunits (green, post-permeabilization). Arrows indicate excitatory synapses where Fib-Tau and α3-NKA/AMPA/NMDA are co-localized (C). Quantification of the fluorescence intensity (indicating size of clusters, see Materials and Methods) of synaptic α3-NKA/AMPA/NMDA spots (obtained after thresholding) following exposure to Fib-Tau showed a reduction in the size of α3-NKA and increase in the size of GluA2 subunit containing AMPA receptors (D). Mann–Whitney test, *n* is number of images analyzed from 3 to 4 experiments (α3-NKA: 75; GluA1: 95; GluA2: 50; GluN1/N2B: 45). Box-plot represents median, interquartile range, and 10–90% distribution, **$P < 0.01$; ns = not significant. Scale bar, 5 μm.

pulled down by Fib-Tau (Fig 4D), that Fib-Tau can be cross-linked to GluA2 (Fig 4D), and that Fib-Tau co-immunoprecipitate with GluA2 (Fig 4E), strongly suggest that the two proteins interact directly. To assess this interaction further, we identified by mass spectrometry the peptide through which GluA2 is cross-linked to Fib-Tau (Appendix Fig S9A–C). GluA2 was found cross-linked through lysine

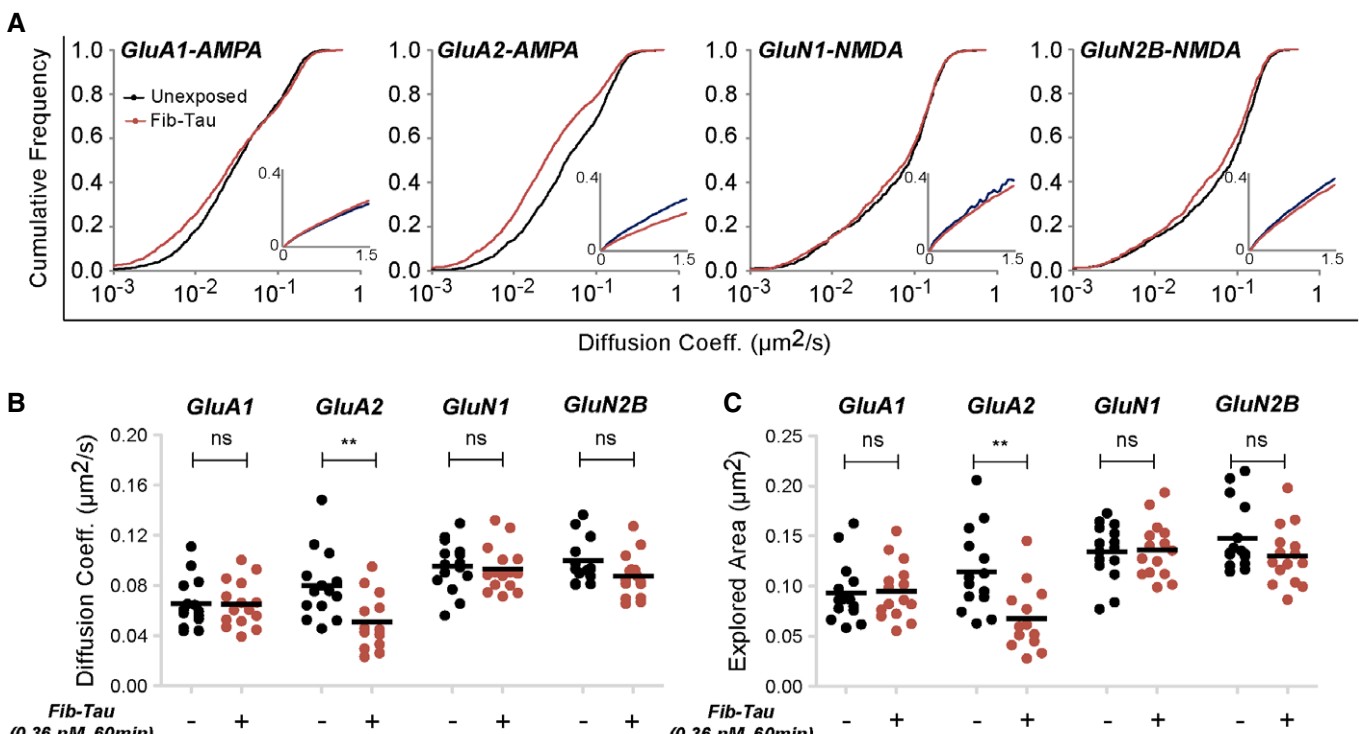

**Figure 6. Diffusion properties of GluA2-AMPA receptors and co-clustering with fibrillar Tau.**

A–C   SPT-QD of pHluorin-tagged AMPA (GluA1 and GluA2) and NMDA (GluN1 and GluN2B) receptor subunits using GFP antibody pre-coupled to QD655. Primary neurons were exposed or not to Fib-Tau (0.36 nM, 60 min). Cumulative frequency of diffusion coefficient of all trajectories (extracted from MSD plot, inset, A) shows a slowdown in diffusion of GluA2 subunit but not GluA1, GluN1, and GluN2B subunits (A). To compute statistical differences, the diffusion coefficient of all trajectories per movie (from three independent experiments) was averaged (B). To measure statistical differences in MSD, the explored area (area under the curve during 600–900 ms of MSD plot) per movie (from three independent experiments) was averaged (C). Decreased diffusion coefficient and area explored for GluA2-AMPA were observed indicating increased confinement concomitant with increased clustering (Fig 5C) following exposure to Fib-Tau. Mann–Whitney test, $n$ is number of movies (left to right): 14, 15, 14, 13, 15, 15, 15, and 15. **$P < 0.01$; ns = not significant.

684, lying within the peptide (SKIAVFDK) spanning GluA2 residues 683–690 (Appendix Fig S9A and B) to Tau-1N3R peptides spanning amino acid residues 142–151, 193–201, 246–257, and 290–309 (Appendix Fig S10). Thus, Fib-Tau binds close to the second extracellular loop of GluA2-AMPA receptor (Appendix Fig S9D). We also performed live-neuron single-particle tracking measurements using quantum dots (SPT-QD; Triller & Choquet, 2008) to confirm that Fib-Tau directly interacts with GluA2 subunit. SPT-QD is a highly sensitive method that allows probing transient and local protein–protein interaction and tracking single molecule for several seconds, in non-invasive imaging conditions (Triller & Choquet, 2008; Choquet & Triller, 2013). The measurements were performed on neurons transfected with pHluorin-tagged (Kopec et al, 2006) GluA1 and GluA2 subunits of AMPA receptors and GluN1 and GluN2B subunits of NMDA receptors using GFP antibody pre-coupled to QD-655. The cumulative frequency of diffusion coefficient of all trajectories (Fig 6A) and averaged diffusion coefficient value per movie is plotted (Fig 6B). The mobility of GluA2-AMPA but not GluA1-AMPA or GluN1-NMDA and GluN2B-NMDA upon exposure of neurons to Fib-Tau (0.36 nM, 60 min) was reduced. The explored area, determined from MSD plot, shows a decrease for GluA2-AMPA (Fig 6C), meaning an increased confinement favoring the notion that slowdown in diffusion is leading to increased clustering in the presence of Fib-Tau.

No changes in intracellular $Ca^{2+}$ concentration were observed upon exposure of mature primary neurons (> DIV 21), loaded with the calcium ($Ca^{2+}$) dye Fluo-4, to Fib-Tau (0.36 nM). The drug 4-aminopyridine (4-AP) readily induces neuronal activity in an action-potential and glutamate receptor-dependent manner (Strowbridge, 1999) as illustrated by $Ca^{2+}$ influx. Neurons exposed to Fib-Tau (0.36 nM, 60 min) exhibited small but significantly higher 4-AP-induced $Ca^{2+}$ response (Fig EV3A and D). In the presence of the $Na^+$-dependent action-potential blocker tetrodotoxin (TTX) or the AMPA receptor antagonist CNQX (6-cyano-7-nitroquinoxaline-2,3-dione), neurons did not respond to the addition of 4-AP (50 μM; Fig EV3B–D). These observations suggest that neurons exposed to Tau fibrils exhibit higher vulnerability due to higher $Ca^{2+}$-influx following action potential. This may be the consequence of increased number of AMPA receptors at synapses.

### Exogenous Fib-Tau destabilizes α3-NKA and affects its turnover

Exogenous Fib-Tau affects the dynamics of α3-NKA and GluA2 subunit of AMPA receptors (Fig 5D). While GluA2-AMPA receptor is enriched in post-synaptic excitatory domains (Greger et al, 2017), α3-NKA is distributed throughout the plasma membrane (Azarias et al, 2013), both at synaptic and non-synaptic sites and also on axons. α3-NKA rapidly extrudes $Na^+$ out of neurons and thus is

responsible for the maintenance of the electrical gradient across the plasma membrane (Azarias *et al*, 2013). Notably, the proportion of Fib-Tau co-localizing with α3-NKA (25–35%) is higher than that of homer (10–20%). We assessed the impact of Fib-Tau interaction with α3-NKA using SPT-PALM (single-particle tracking using photoactivated localization microscopy). SPT-PALM allows studying simultaneously the dynamics of thousands of single molecules with higher accuracy (Shrivastava *et al*, 2013b). PALM imaging was performed on DIV 21 neurons, transfected on DIV 18 with α3-NKA-plasmid tagged with Dendra2. Transfected dendrites/shafts were identified by epifluorescence (green, Fig 7A). Single molecule trajectories (minimum 5-point length) were recorded and shown. The diffusion coefficient of α3-NKA in shafts, derived from the initial slope of MSD curve (Fig 7B), increased upon exposure of neurons to Fib-Tau (0.36 nM, 60 min; Fig 7C). The MSD plot (Fig 7B) also revealed that exogenous Fib-Tau reduced the confinement (upward bent of curve) within dendritic shaft. This suggests that exogenous Tau fibrils trigger the relocation of α3-NKA outside the synapses. α3-NKA distribution and dynamics were unaffected by Fib-Tau fibrils in spines.

The biophysical property of α3-NKA on neuronal membrane is not fully documented. A membrane protein can have multiple diffusive states that depend on their interactions with partner proteins (Persson *et al*, 2013). Here, two diffusive states of α3-NKA were identified with confidence in neurons exposed or not to Fib-Tau using minimum 2-point long trajectories. α3-NKA was found to populate a *Bound State* (slow-diffusing; assumed bound to native partner/scaffold proteins) and a *Free State* (fast-diffusing; assumed unbound and freely diffusing; Fig 7D). The proportion of α3-NKA in each state was derived from SPT-PALM trajectories measurements using recently described variational Bayesian treatment of hidden Markov models that combine information from thousands of short single-molecule trajectories (Persson *et al*, 2013). The proportion of α3-NKA molecules in the *Free State* remained unchanged upon exposure of neurons to Tau fibrils while that in the *Bound State* decreased from 71.4 to 57.1% (Fig 7D). Importantly, the occupancy time, i.e., the percentage of time α3-NKA trajectories spent in *Bound State* (Fig 7E), and the dwell time of trajectories in *Bound State* (Fig 7F) decreased

significantly upon exposure of neurons to Fib-Tau. Conversely, the occupancy time in *Free State* increased slightly in the presence of Fib-Tau. These observations suggest that Fib-Tau destabilizes native, functional, sodium–potassium pump on the plasma membrane. To assess to what extent α3-NKA destabilization is specific to Fib-Tau, we repeated SPT-PALM experiments following exposure of neurons to monomeric Tau (50 nM). Monomeric Tau was rapidly internalized by neurons as it was detected in the cytosolic compartments after a 60-min exposure (Fig EV4A). Monomeric Tau did not alter α3-NKA diffusion coefficient (Fig EV4B), occupancy in bound/free state (Fig EV4C), or dwell time in bound/free state (Fig EV4D).

To assess the consequences of NKA interaction with Fib-Tau, we performed cell-surface biotinylation and compared the total amount of NKA exposed at the plasma membrane in the absence or the presence of exogenous Fib-Tau (1.8 nM, 60 min). The amount of NKA exposed at the surface of neurons decreased by 30% upon exposure of neurons to Fib-Tau (Fig 7G and H). To understand the reason of NKA decrease, neurons expressing pHluorin-tagged α3-NKA were exposed to ATTO-647-labeled Fib-Tau and the cells were imaged. α3-NKA was found within spots, reminiscent of endocytic vesicles (Fig 7I, green, top row; Fig 7I, green, bottom row) after fixation with paraformaldehyde. Fib-Tau clusters co-localized perfectly with α3-NKA spots (Fig 7I, red, top row, arrows). To demonstrate that α3-NKA spots are related to early endocytotic pathway, neurons were co-transfected with α3-NKA-pHluorin (green) and clathrin-mRFP (blue; Tagawa *et al*, 2005) at DIV 18 and exposed to ATTO-647-Fib-Tau (red) on DIV 21 (Fig 7J). Most α3-NKA spots co-localizing with Fib-Tau were found overlapping with clathrin pits (Fig 7J). Reduced cell-surface levels of NKA may decrease neuron capacity to restore $Na^+$ level after depolarization (Azarias *et al*, 2013; Shrivastava *et al*, 2015). Sodium imaging (see Materials and Methods) revealed that neurons exposed to Fib-Tau (0.36 nM, 60 min) exhibit reduced rate of $Na^+$ extrusion (Fig EV5). This observation suggests that α3-NKA activity is compromised upon exposure of neurons to Fib-Tau as it is responsible for the initial rapid clearance of $Na^+$ from neurons following depolarization.

▶

**Figure 7. Exogenous Fib-Tau destabilizes α3-NKA and affects their turnover.**

A   Single-particle tracking using PALM (SPT-PALM) was performed on α3-NKA-Dendra expressing primary neurons. A representative image shows an epifluorescence image and trajectories (strands of different colors) of minimum 5 points in length.

B, C   Extraction of diffusion coefficient using mean squared displacement (MSD) approach using minimum 5-point trajectory length. Spine and shaft were visually identified. An upward shift of MSD curve, indicating reduced confinement, was observed following exposure to Fib-Tau (0.36 nM, 60 min) (B). As expected, an increased diffusion coefficient (i.e., increased mobility) (C) of α3-NKA is observed in dendritic shaft following exposure to Fib-Tau. Mann–Whitney test, *n* = 21 cells, obtained from four independent experiments; each dot represents average diffusion coefficient of all trajectories value per cell.

D–F   Extraction of the number of diffusive states of α3-NKA and the transition rates between these diffusive states using hidden Markov model (see Materials and Methods) using thousands of short (minimum 2-point) trajectories. Two diffusive states could be derived for α3-NKA with confidence (termed bound/slow-diffusing or free/fast-diffusing) (D) between which α3-NKA can transition. The characteristic parameters for neurons exposed or not to Fib-Tau are shown (D). The proportion of single molecules transitioning to bound state decreased (from 71.4 to 57.1%) following neuron exposure to Fib-Tau. The occupancy time of α3-NKA in bound state also decreased whereas that in free state increased upon neuron exposure to Fib-Tau (E). Dwell time of α3-NKA is also reduced in bound state indicating destabilization of native α3-NKA-complexes following interaction with Fib-Tau (F). Box-plot represents median, interquartile range, and entire distribution, Mann–Whitney test on averaged value per movie from four experiments (*n* = 21 cells).

G, H   Cell-surface biotinylation on primary neurons exposed or not to Fib-Tau (1.8 nM, 60 min). The amount of α-NKA at cell surface decreased upon exposure of neurons to Fib-Tau (G). Quantification was performed by normalizing the Western blot band intensity with input. Paired *t*-test, three independent experiments.

I   Paraformaldehyde fixation α3-NKA-pHluorin (green, top row) reveals endocytic spots otherwise not visible in live neurons (green, bottom row). Many Fib-Tau-ATTO-647 (red) clusters overlapped (arrow) with α3-NKA-pHluorin spots (exposure, 0.36 nM, 60 min).

J   The majority of the overlapping Fib-Tau-ATTO-647 clusters (red) and α3-NKA-pHluorin spots (green) localized on top of clathrin-mRFP (blue) puncta.

Data information: *$P < 0.05$; ns = not significant. Scale bars: 1 μm in panel (A) and 2 μm in panels (I and J).

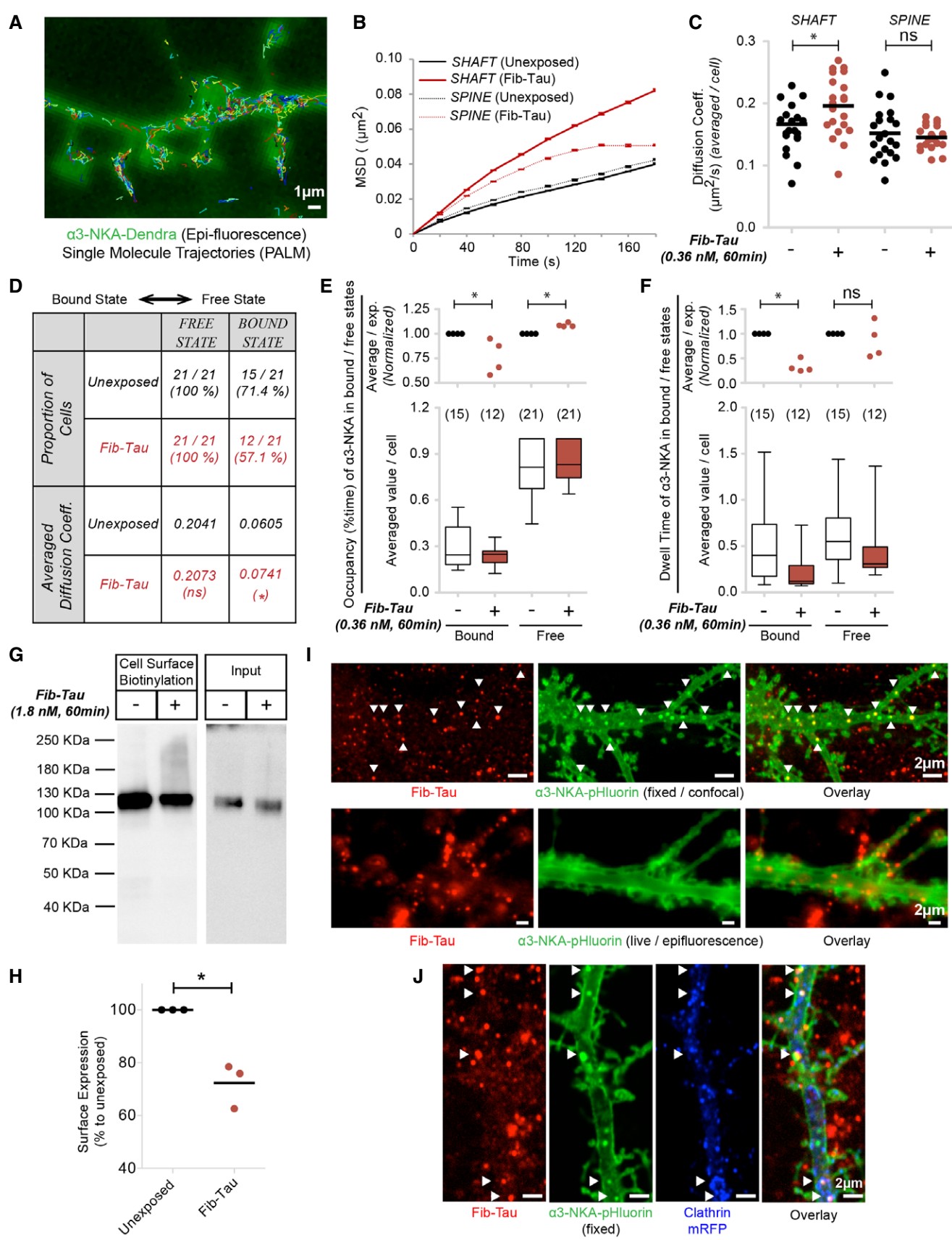

**Figure 7.**

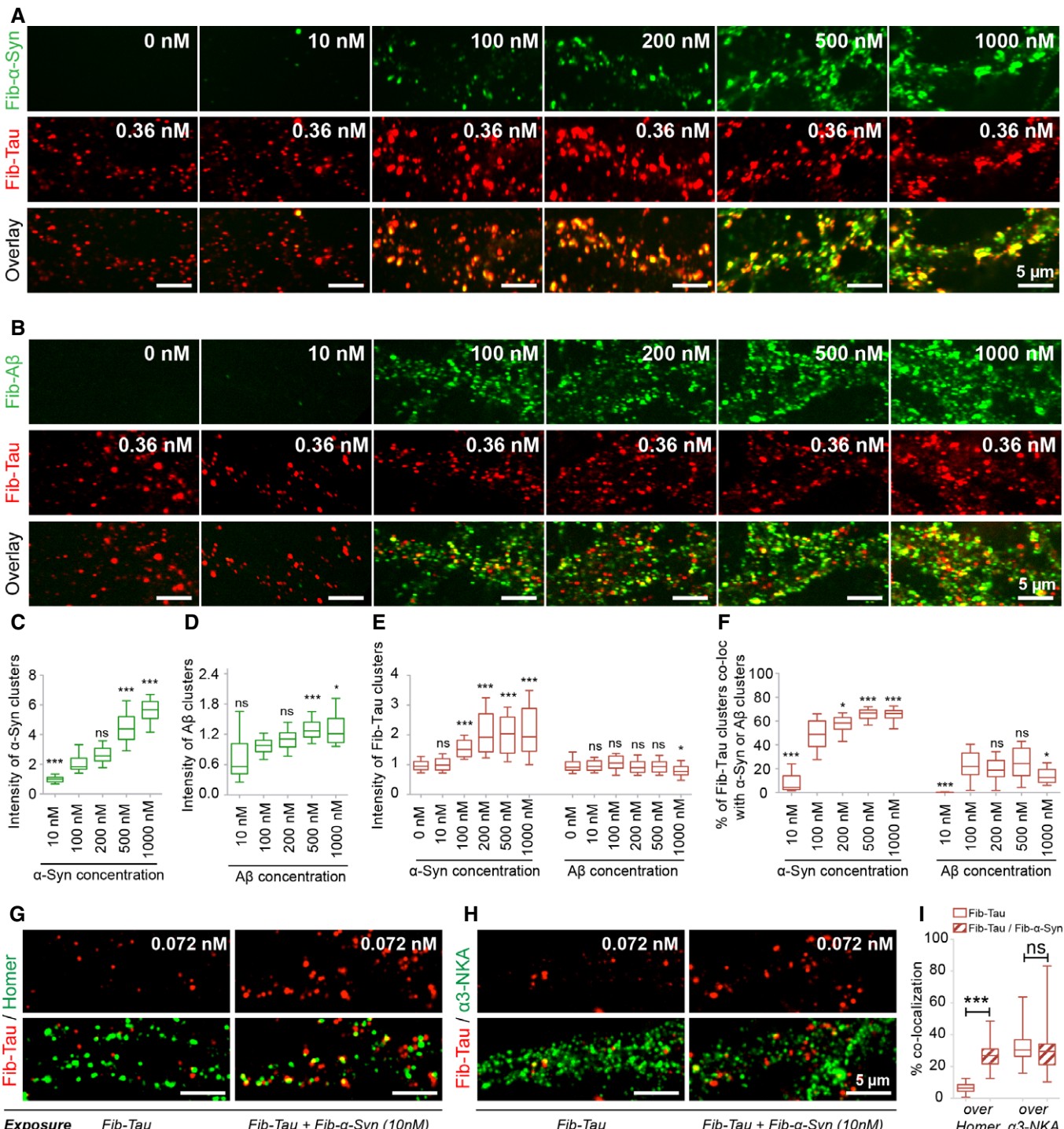

**Figure 8.**

## α-Synuclein but not amyloid-β fibrils enhance fibrillar Tau clustering within neuron membranes

Aβ, α-Syn, and Tau deposits often co-exist in various neurodegenerative disorders (Irwin *et al*, 2013; Morales *et al*, 2013). As all of these assemblies spread from one cell to another, it is reasonable to postulate that the binding of one fibrillar protein assembly to neurons may interfere with that of another and affect its diffusion and clustering within neuronal membrane. We therefore documented the potential cross-talk between Tau, Aβ, and α-Syn fibrils. Neurons were exposed to ATTO-550-labeled Fib-Tau (0.36 nM, 60 min) and increasing concentration of α-Syn (Fig 8A) or Aβ fibrils (Fig 8B; 10–1,000 nM, monomer-equivalent concentrations, labeled with ATTO-647 or not). A concentration-dependent increase in the clustering of α-Syn fibrils was observed (Fig 8A and C), in agreement with previous findings (Shrivastava *et al*,

◄

**Figure 8.  α-synuclein but not amyloid-β fibrils enhance fibrillar Tau clustering within neuron membranes.**

A–F  DIV 21 neurons were co-exposed to fixed concentration of Fib-Tau (0.36 nM, 60 min, ATTO-550 labeled, red) and increasing concentrations (given in monomer-equivalent) of α-Syn (A) or Aβ (B) fibrils [in total, five experiments were performed with either ATTO-647-labeled α-Syn or Aβ (shown here in green) or unlabeled α-Syn or Aβ fibrils]. A concentration-dependent increase in the binding and clustering of both α-Syn fibrils (A, C) and Aβ fibrils (B, D) is seen. One-way ANOVA with Dunnett's *post hoc* test was performed to compare the different condition to the reference condition (100 nM). The number of images analyzed is 27, 25, 25, 26, and 28 in (C) and 18, 50, 50, 50, and 25 in (D), from left to right. The data originate from 3 to 4 independent experiments. Notably, Fib-Tau clustering dramatically increased with increasing concentrations of α-Syn, but not Aβ fibrils (A, B, E). The quantification in (E) includes all five experiments (*n* = number of images analyzed, left to right: 58, 54, 50, 62, 63, 65, 43, 35, 60, 60, 60, 35; one-way ANOVA with Dunnett's *post hoc* test was performed to compare the size of Fib-Tau clusters). A perfect co-localization between Fib-Tau and α-Syn clusters is also observed. The proportion of Fib-Tau clusters co-localizing with α-Syn fibrils increased with an increasing concentration of α-Syn fibrils reaching 60–70% overlap (A, F). A weaker, concentration independent, overlap (10–20%) was observed with Aβ fibrils (B, F). The quantification in (F) is derived from 3 to 4 independent experiments. The number of images analyzed is 27, 25, 25, 26, and 28 for α-Syn and 18, 50, 50, 50, and 25 for Aβ from left to right. One-way ANOVA with Dunnett's *post hoc* test was performed to compare the co-localization differences from 100 nM concentration. Box-plot represents median, interquartile range, and 10–90% distribution.

G–I  Primary neurons were co-exposed to Fib-Tau (0.36 nM, ATTO-550 labeled) and Fib-α-Syn (50 nM, monomer-equivalent, unlabeled) for 60 min and immuno-labeled for homer (G) or α3-NKA (H). Quantification of Fib-Tau clusters co-localized with homer or α3-NKA (I). Note an increased co-localization of Fib-Tau with homer in the presence of Fib-α-Syn. Box-plot represents median, interquartile range, and 10–90% distribution; Mann–Whitney test was performed, *n* = 40 images from four experiments.

Data information: Scale bars, 5 μm. *$P < 0.05$, ***$P < 0.001$; ns = not significant.

---

2015). A concentration-dependent increase in the clustering of Aβ fibrils was also observed (Fig 8A and D), although to a lesser extent than that of α-Syn fibrils or Aβ oligomers (Renner *et al*, 2010; Shrivastava *et al*, 2013a). Fib-Tau clustering increased significantly with increasing concentration of α-Syn fibrils but not Aβ fibrils (Fig 8E). Enhanced Fib-Tau clustering correlates with the co-clustering of Tau and α-Syn fibrils (Fig 8A and F). No significant co-clustering was observed upon addition of increasing amounts of Aβ fibrils. α-Syn fibrils did not affect the distribution of monomeric Tau (Fig EV4E–H).

A proportion of Fib-Tau clusters at excitatory synapses where they alter the composition of α3-NKA and AMPA receptors (Fig 5). To identify the localization of Tau clusters, we exposed neurons with a very low concentration of fibrils (10 nM monomer-equivalent, 60 min) followed by immunodetection of homer. The proportion of Fib-Tau clusters at synapses increased from 6.4 to 27.5% in the presence of Fib-α-Syn (Fig 8G and I). At the same time, the co-localization (30%) between Fib-Tau and α3-NKA was unaltered, the latter being localized both in and out of synapses (Fig 8H and I). These observations strongly suggest a cross-talk between Tau and α-Syn fibrils upon binding to neuronal membrane.

# Discussion

The spread of aggregated Tau protein particles is believed to contribute to the progression of tauopathies, including Alzheimer's disease. Several laboratories have successfully demonstrated that Tau high molecular weight assemblies exhibit prion-like properties following injection in the brain of rodents (Clavaguera *et al*, 2009; Dujardin *et al*, 2014; Sanders *et al*, 2014; Iba *et al*, 2015; Takeda *et al*, 2015; Guo *et al*, 2016; Narasimhan *et al*, 2017). Here, we describe the early events (minutes to hours) occurring on the cell surface of neurons prior to the endocytosis of exogenous Tau fibrils. We demonstrate that fibrils made of Tau isoforms 1N3R or 1N4R bind to neuron plasma membrane, diffuse laterally with time, and form clusters at excitatory synapses. To identify fibrillar Tau neuron membrane partners, we exposed neurons briefly to fibrillar Tau, pulled down on the fibrils, and identified the associated proteins by mass spectrometry. Twenty-nine partner membrane proteins with extracellular domains were found to interact with both Tau 1N3R

and 1N4R fibrils. Of particular interest are NKA complex α1, α3 and β1 subunits, AMPA receptor GluA1 and GluA2 subunits, and NMDA receptor GluN1 and GluN2B subunits. First, these proteins could be cross-linked to the fibrils and second, fibrillar Tau co-immunoprecipitates with these proteins. Thus, they are likely to interact directly with fibrillar Tau.

## Binding and clustering of Fib-Tau within neuronal plasma membrane

Pathogenic proteins form clusters that act as traps for membrane proteins diffusing at the plasma membrane (Renner *et al*, 2010; Shrivastava *et al*, 2013a, 2015, 2017; Goniotaki *et al*, 2017). Similarly to Aβ oligomers and α-Syn oligomers/fibrils (Renner *et al*, 2010; Shrivastava *et al*, 2013a, 2015), exogenous Tau fibrils clustered at the plasma membrane in a concentration- and time-dependent manner. The finding that Fib-Tau diffusion is confined after 60 min most probably reflects a time-dependent increased surface crowding. In addition, plasma membrane favors clustering of Fib-Tau due to inter-molecular interactions in a crowded environment (Shrivastava *et al*, 2017). Thus, the higher the concentration of Fib-Tau, the more clusters will be detected, in agreement with our observations. In contrast to Aβ oligomers and α-Syn oligomers/fibrils, Fib-Tau clustering was slow and the cluster number decreased rapidly (within an hour) after removal of the fibrils from the culture medium. Furthermore, we observed that Tau fibrils are dynamic at the plasma membrane and that the clusters are unstable with at most 50% of Tau fibrils within clusters. In contrast, at similar concentrations, 90% of α-Syn fibrils reside within clusters at any time point (Shrivastava *et al*, 2015). These observations suggest that Fib-Tau clusters either (i) disassemble and diffuse at the plasma membrane, or (ii) detach from the membrane, or (iii) are endocytosed and processed within the cells (Holmes *et al*, 2013; Woerman *et al*, 2016; Flavin *et al*, 2017).

## Interaction between α3 NKA and GluA2 and exogenous Tau fibrils

The clustering of Fib-Tau at excitatory synapses alters synaptic protein composition with a reduction in the total amount of synaptic NKA α3 subunit and an increase in the amount of AMPA receptor GluA2 subunit. NMDA receptor subunit distribution and amount

were overall unaffected by Fib-Tau clustering within excitatory synapses.

The binding of Aβ oligomers to AMPA receptors has been reported (Zhao *et al*, 2010). α-Syn is also likely to interact with AMPA receptors (Hüls *et al*, 2011). SPT-QD measurements revealed that the binding and clustering of Fib-Tau reduce AMPA receptor GluA2 subunit diffusion and increases their confinement at excitatory synapses. Interestingly, we found that neurons exposed to Fib-Tau were more vulnerable following action-potential-induced $Ca^{2+}$-influx. This may be the consequence of AMPA receptor redistribution and confinement at excitatory synapses. Importantly, however, since Fib-Tau clusters are relatively unstable, the diffusion trapping of GluA2-AMPA receptor subunit is likely to be transient, leading to an initial stimulus-evoked calcium influx. Thus, the interaction of AMPA receptors with Fib-Tau clusters at neuron plasma membrane may lead to the reorganization of AMPA/NMDA receptors depending on local calcium levels (Choquet & Triller, 2013).

We found that NKA pump α1, α3, and β1 subunits interact with Tau fibrils. The minimal functional unit of NKA is either α3β1 or α1β1 complexes, the α3β1 complex being exclusively expressed in neurons. NKA α3 subunit was recently reported to interact with extracellular amylospheroids (Ohnishi *et al*, 2015) and α-Syn oligomers/fibrils (Shrivastava *et al*, 2015). While α-Syn fibril clusters acted as diffusion trap for NKA α3 subunit leading to compromised NKA activity (Shrivastava *et al*, 2015, 2017), Fib-Tau clusters destabilized NKA. The amount of NKA α3 subunit exposed to the surface of the cells decreases significantly with concomitant formation of NKA α3 subunit containing spots, compatible with endocytic vesicles. Fib-Tau clusters were present in most NKA α3 subunit containing spots and the spots overlapped with clathrin immunoreactivity. Altogether, these observations suggest that Fib-Tau clusters destabilize NKA complexes, thus, reducing neuron's capacity to control membrane depolarization (Azarias *et al*, 2013). It is likely that the displaced NKA α3 subunit is endocytosed together with Fib-Tau clusters by neurons.

The precise role of neuron membrane proteins such as glypican, the glycoprotein M6, neurexin, and the voltage-dependent anion selective channel proteins that interact with Fib-Tau needs further evaluation since these membrane proteins also interact with pathogenic α-Syn assemblies (Shrivastava *et al*, 2015). While they may play a role in endocytosis of Fib-Tau (Holmes *et al*, 2013; ref in Shrivastava *et al*, 2017), they may have additional functions. Overall, several membrane protein complexes with extracellular domains interact with both Fib-Tau, pathogenic α-Syn assemblies, and oligomeric Aβ (Shrivastava *et al*, 2017). Such events are key for pathogenic protein assemblies-mediated signaling impairment and prion-like propagation.

Altogether, our observations suggest that Fib-Tau interact with the neuron plasma membrane via numerous protein partners, critical for neuronal activity, as opposed to a unique receptor. This is reminiscent of pathogenic α-Syn assemblies (Shrivastava *et al*, 2017). This has implications on therapeutic strategies targeting the binding of pathogenic assemblies to neuronal membrane. Interfering with binding will require either the design of ligands that change the surface properties of fibrils (Melki, 2017) or shielding not one but multiple partner proteins. Thus, luring pathogenic fibrils to avoid the redistribution of membrane protein partners will require not one but multiple peptide decoys, reproducing the protein partners' interaction sites. Further studies may allow establishing a hierarchy among fibrillar Tau partners and defining protein interactions amenable to inhibition in view of therapeutic interventions.

## Cross-talk between α-synuclein, amyloid-β, and Tau fibrils at neuron plasma membrane

Several reports highlight the coexistence of Aβ, α-Syn, and Tau deposits in the brain of patients at late stages of PD and AD (Compta *et al*, 2011; Howlett *et al*, 2015; Irwin *et al*, 2017). This is certainly due to proteostasis collapse (Labbadia & Morimoto, 2015). It is reasonable to envisage that the propagation of one pathogenic protein assembly interfere with that of others at early stages of disease. Here, we documented the potential cross-talk between Tau, Aβ, and α-Syn fibrils. We show that α-Syn but not Aβ fibrils enhance Fib-Tau clustering on the plasma membrane. Interestingly, we found that Tau fibrils co-cluster with pathogenic α-Syn fibrils at excitatory synapses in hippocampal neurons. This suggests that fibrillar α-Syn-mediated clustering of partner proteins within neuronal plasma membrane yields structures highly populated in partner proteins common to fibrillar α-Syn and Tau. Such pathogenic protein scaffolds will favor Fib-Tau binding. Thus, our observations suggest that the prion-like propagation of α-Syn fibrils within the brain of PD patients may facilitate that of Fib-Tau. Altogether, our results support the concept that neuron plasma membrane plays the role of a chemical reactor allowing the clustering of pathogenic assemblies (Shrivastava *et al*, 2017).

# Materials and Methods

### Generation, labeling, and characterization of Fib-Tau, α-Syn, and Aβ

Recombinant wild-type and full-length human Tau-1N3R and 1N4R Tau were expressed and purified as described previously (Tardivel *et al*, 2016). Tau protein concentration was determined by spectrophotometry using an extinction coefficient at 280 nm of 7,450 per M/cm. Aliquots of pure Tau proteins at concentrations of 50–100 μM were stored at −80°C. Fibrillation of Tau1N3R and 1N4R was achieved at 40 μM in the presence of 10 μM heparin by shaking 0.5 ml solution aliquots at 37°C in an Eppendorf Thermomixer set at 600 rpm for 4 days. At steady state, fibrils were spun for 35 min at 20°C and $113,000 \times g$. The amount of Fib-Tau was estimated by subtraction of the soluble fraction remaining after centrifugation from the initial concentration. The pelleted material was resuspended in PBS at an equivalent monomeric Tau1N3R or 1N4R concentration of 40 μM.

### Biotin labeling

Labeling of Fib-Tau-1N3R and 1N4R was achieved by the addition of 10 molar equivalents of EZ-Link-Sulfo-NHS-S-S-Biotin (Thermo Scientific #2133) for 1 h at room temperature. The reaction was stopped by addition of Tris–HCl buffer pH7.5 to a final concentration of 40 mM and incubation during 10 min.

### ATTO labeling

Labeling of Fib-Tau was achieved by the addition of 2 molar equivalents of lysine-reactive ATTO-488, ATTO-550, or ATTO-647N (ATTO-TEC, GMBH) for 1 h at room temperature. The unreacted fluorophore was removed by two cycles of centrifugation at 15,000 *g* for 10 min and resuspension of the pellet in PBS. Representative transmission electron micrographs of Fib-Tau used throughout this study are presented in Appendix Figs S1 and S2.

The amount of ATTO-488, ATTO-550, and ATTO-647N incorporated within Fib-Tau was assessed by mass spectrometry. Protein samples at 40 μM in PBS were diluted 1:5 to 1:10 (v/v) in 1% trifluoroacetic acid. Diluted protein sample was further mixed volume to volume with α-cyano-4-hydroxycinnamic acid (4HCCA) matrix solution at 20 mg/ml in 70% acetonitrile and 5% formic acid. Protein-matrix solution (0.5–1 μl) was spotted on a stainless steel MALDI target (Opti-TOF; Applied Biosystems). 4HCCA matrix was chosen for its ability to produce a high number of charge states of proteins. Mass spectra were acquired with a MALDI-TOF/TOF 5800 mass spectrometer (Applied Biosystems) using linear mode acquisition. External calibration was performed using singly, doubly, and triply charged ions of unmodified WT (wild-type) human α-synuclein and four and five charged ions of unmodified WT 1N3R and 1N4R Tau proteins. Experimental measured molecular masses of Tau proteins were calculated using their tetra- and pentacharged ions. Acquisition and data analysis were performed using the Data Explorer software from Applied Biosystems. The MALDI-TOF mass spectra of unlabeled, ATTO-550, ATTO-488, ATTO-647N, and biotin-labeled Fib-Tau-1N3R fibrils (from top to bottom, left column) and Fib-Tau-1N4R (from top to bottom, right column) are shown in Appendix Fig S2. The spectra show that Fib-Tau-1N3R and 1N4R molecules within fibrils are labeled by 1 to 2 ATTO and 3 to 4 biotin molecules on average.

To obtain fibrils homogeneous in size, labeled Fib-Tau-1N3R and 1N4R were fragmented for 5 min in 2 ml Eppendorf tubes in a VialTweeter powered by an ultrasonic processor UIS250v (250 w, 24 kHz, Hielscher Ultrasonic, Teltow, Germany) set at 75% amplitude, 0.5 s pulses every 1 s. The morphology of Fib-Tau was assessed by TEM in a Jeol 1400 transmission electron microscope following adsorption onto carbon-coated 200 mesh grids and negative staining with 1% uranyl acetate. The images were recorded with a Gatan Orius CCD camera (Gatan).

The expression and purification of human wild-type α-Syn were performed as previously described (Ghee *et al*, 2005). α-Syn fibrils were assembled and labeled as described (Bousset *et al*, 2013). The expression and purification of Met-Aβ 1–42 were performed as described (Walsh *et al*, 2009). Aβ 1–42 was assembled in DMEM, at 37°C without shaking for 24 h.

### Calculation of Fib-Tau particle concentration

Fib-Tau particle concentration for a given equivalent monomeric Tau concentration was determined using analytical ultracentrifugation and quantitative transmission electron microscopy.

Sedimentation velocity measurements were carried out using a Beckman Optima XL-A ultracentrifuge equipped with UV-visible detection system using an AN60-Ti four-hole rotor and cells with two-channel 12-mm pathlength centerpieces. For Fib-Tau, the samples (400 μl in PBS at 40 μM equivalent monomeric Tau) were

spun at 16,364 × *g* and 20°C. Sample displacement profiles were obtained by recording the absorbance at 280 nm every 5 min. The sedimentation coefficient distributions of fragmented Fib-Tau 1N3R and 1N4R were obtained using the least-squares boundary modeling ls-g*(s) using the software Sedfit (National Institutes of Health, Bethesda, MD; Schuck, 2000; Schuck *et al*, 2002). The buffer density (1.0053 g/ml), viscosity (1.0188 cp), and f/f0 (1.7) were calculated with the software Sednterp. For all the measurements, the sedimentation coefficient values were corrected to s20,w (standard solvent conditions in water at 20°C). The sedimentation boundaries obtained at intervals of 10 min are shown (Appendix Fig S1A). The ls-g*(s) distribution of Fib-Tau 1N3R (Appendix Fig S1B) revealed the presence of particles with sedimentation coefficients centered at 66S (s20,w = 68.4 S), corresponding to particles of ~5,442 kDa, i.e., made of ~137 monomers. These measurements allowed us to determine the average particle concentration of Fib-Tau corresponding to a given equivalent monomer concentration. At a working concentration of 1 μM, the particle concentration of Fib-Tau is 1 μM/137 = 0.007 μM.

Tau fibrils were spun in an Airfuge EM90 rotor (Beckman Instruments, Inc., Brea, CA) onto carbon-coated 200 mesh grids and negatively stained with 1% uranyl acetate. The images were recorded with a Gatan Orius CCD camera (Gatan). An example is displayed (Appendix Fig S1C). The length distribution graph of the fibrils (Appendix Fig S1D) was derived from measurement of the length of 838 fibrils imaged in three quantitative negatively stained electron micrographs. The graph shows that Fib-Tau is on average 35 nm long. The average diameter was measured (18.7 nm). The fibril volume was calculated: $3.1415 * radius^2 * length = 9.51 \times 10^{-24}$ m$^3$. The volume of an average amino acid residue with a molecular mass 110 is $1.28 \times 10^{-28}$ m$^3$ and Tau-1N3R is composed of 381 amino acid residues and their water content is 30%. Thus, the number of Tau monomers within the fibrils used throughout this study is 136 in perfect agreement with the number derived from analytical ultracentrifugation measurements.

### Pull-down of biotinylated Fib-Tau and preparation of pulled down protein samples

Biotinylated Fib-Tau-1N3R or 1N4R was applied at 14.4 nM (particle concentration) in the culture medium of 2-week-old pure cortical mouse neuron cultures (10 culture dishes, 60-mm, per fibril type). Unexposed neurons grown under the same conditions were used as control. After 10 min, cells were washed twice with 1× PBS and scraped on ice in Lysis buffer (50 mM Tris–HCl pH 7.5, 2 mM EDTA, 0.1% Triton X-100), supplemented with complete protease inhibitor cocktail (Roche; 500 μl per dish). The resulting solutions were flash frozen in liquid nitrogen and stored at −80°C. Cell lysis was completed by sonication, and cell debris was removed by centrifugation at 500 × *g*, 5 min at 4°C. Protein concentration in the extracts was determined using the BCA assay kit (Thermo Scientific, Waltham, MA). To pull down biotin-labeled Fib-Tau-1N3R and 1N4R together with their partner proteins, 100 μl streptavidin magnetic beads (Pierce, Waltham, MA) were washed three times in binding buffer (50 mM Tris–HCl pH 7.5, 150 mM NaCl, 0.1% Triton X-100, complete protease inhibitor cocktail). Neuronal lysates (1 mg total protein) were added to the settled streptavidin magnetic beads and incubated for 1 h at 4°C under gentle agitation. Control,

unexposed neuron lysates were also incubated with streptavidin magnetic beads. After three washes with 500 μl of binding buffer, the proteins bound to the settled beads (25 μl) were denatured by addition of 25 μl of Laemmli buffer at 95°C and resolved on a 10% SDS–PAGE gel, for concentration purpose into a single band of about 5 mm through a short electrophoretic step. The gels were stained with Coomassie blue, and the protein bands were cut off into unique bands of about 5 mm. Individual bands were subjected to in-gel trypsin digestion as described previously (Shevchenko *et al*, 1996), using standard conditions including reduction and alkylation and overnight digestion with Trypsin Gold (Promega) at a concentration of 10 ng/μl in 25 mM ammonium bicarbonate. Tryptic peptides extracted by addition of one volume of 60% acetonitrile in 0.1 formic acid were vacuum-dried and resuspended in 0.1% (v/v) formic acid prior to nanoLC-MS/MS mass spectrometry analyses.

In the cross-linking experiments, DTSSP (Thermo Scientific, Waltham, MA) was used to cross-link biotinylated Fib-Tau, to extracellularly exposed protein partners. For chemical cross-linking, neurons exposed for 10 min to biotinylated Fib-Tau (14.4 nM equivalent particle concentration) were washed twice with 3 ml artificial cerebrospinal fluid (ACSF, pH 7.4, 5 mM Hepes, 125 mM NaCl, 2.5 mM KCl, 2 mM $CaCl_2$, 1 mM $MgCl_2$, 33 mM glucose). Cross-linking was performed by incubating neurons in 1 ml ACSF containing 1 mM DTSSP for 20 min at 37°C. The cross-linking reactions were stopped by addition of 20 mM Tris–HCl pH 7.5 and further incubation for 5 min at room temperature. The neurons were then scraped in 50 mM Tris–HCl pH 7.5, 150 mM NaCl, 2 mM EDTA, 1% Triton X-100, 0.5% sodium deoxycholate, and complete protease inhibitor cocktail. Tau pull-down, preparation of pulled down protein samples and MS analysis and identification of proteins cross-linked to Fib-Tau were performed as described for non-cross-linked samples.

**Co-immunoprecipitation of Fib-Tau with protein partners**

Extracts (1 mg total proteins) from neurons exposed 10 min to 14.4 nM biotinylated Fib-Tau-1N3R were pre-incubated with protein G magnetic beads (Thermo Scientific; 50 μl beads equilibrated in binding buffer) for 1 h at 4°C under gentle agitation. Then, the pre-cleared extracts (0.25 mg initial total proteins) were incubated overnight at 4°C under gentle agitation in the presence of 1 μg of goat polyclonal anti-$Na^+/K^+$-ATPase alpha3 (C-16, Santa Cruz, # sc-16052), rabbit polyclonal anti-GluA2 (Synaptic Systems, # 182103), or rabbit polyclonal anti-GluN1 (Synaptic Systems, #114011). As controls, the same amounts of extracts were incubated with 1 μg of pre-immune goat or rabbit control IgGs (Santa Cruz, # sc-2028 and # sc-2027, respectively). The day after, samples were incubated with protein G magnetic beads (20 μl equilibrated in binding buffer) for 1 h at 4°C under gentle agitation. After three washes with 500 μl of binding buffer, the protein G-bound protein complexes were eluted by heating at 95°C for 5 min in 20 μl of Laemmli buffer without β-mercaptoethanol. 5 μl of denatured samples was either processed for SDS–PAGE and Western blot analysis or loaded onto nitrocellulose membranes (GE Healthcare) using a 48-slot slot blot filtration apparatus (GE Healthcare). Blots were probed with high-sensitivity streptavidin-HRP (Thermo Scientific) to detect the presence of biotinylated Fib-Tau in the immunoprecipitated samples.

**Mass spectrometric identification and quantification of the pulled down proteins**

Tryptic peptides from six independent replicates were analyzed separately by nanoLC-MS/MS with the Triple-TOF 4600 mass spectrometer (ABSciex) coupled to the nanoRSLC ultra-performance liquid chromatography (UPLC) system (Thermo Scientific) equipped with a trap column (Acclaim PepMap100C18, 75 μm i.d. × 2 cm, 3 μm) and an analytical column (Acclaim PepMap RSLC C18, 75 μm i.d. × 25 cm, 2 μm, 100 Å). Peptides were loaded at 5 μl/min with 0.05% TFA in 5% acetonitrile, and peptide separation was performed at a flow rate of 300 nl/min with a 5–35% solvent B gradient in 120 min. Solvent A was 0.1% formic acid in water, and solvent B was 0.1% formic acid in 100% acetonitrile. NanoLC-MS/MS experiments were conducted in a data-dependent acquisition method by selecting the 20 most intense precursor ions, with charge states 2–5 and above an intensity threshold of 100, for CID fragmentation. NanoLC-MS/MS raw data were processed using the MS Data Converter software (AB Sciex) for generating mgf data files.

Database searching for protein identifications was performed using the Mascot search engine (Matrix Science, London, UK; version 2.4.1) against the SwissProt database selected for Rodentia (SwissProt_2017_06 database release), including a decoy database search. Mascot was searched with a fragment ion mass tolerance of 0.05 Da and a parent ion tolerance of 25 ppm. Carbamidomethylation of cysteine was specified in Mascot as a fixed modification. Oxidation of methionine, Glu->pyro-Glu of the N-terminus, Gln->pyro-Glu of the N-terminus, and acetylation of the N-terminus were specified in Mascot as variable modifications.

Scaffold (version Scaffold_4.5.0, Proteome Software Inc., Portland, OR) was used to validate MS/MS-based peptide and protein identifications. Peptide identifications were accepted if they could be established at greater than 95.0% probability by the Scaffold Local FDR algorithm. Protein identifications were accepted if they could be established at greater than 99.0% probability and contained at least two identified peptides. Protein probabilities were assigned by the Protein Prophet algorithm (Nesvizhskii *et al*, 2003). Proteins that contained similar peptides and could not be differentiated based on MS/MS analysis alone were grouped to satisfy the principles of parsimony.

Identified proteins were quantified by a label-free proteomic approach using spectral counting (Liu *et al*, 2004) with the Scaffold software. Only proteins identified with at least two unique peptides in at least four replicates out of six were quantified. Only proteins with a total spectral count ratio, between the cells exposed to Fib-Tau and the control cells, above 2 and a *P*-value < 0.05 were considered as significantly increased in the pull-down and thus considered as proteins interacting with Fib-Tau. Total spectra count ratios presented in Figs 4D and EV1C, and Appendix Tables S1–S3 were calculated from averaged total spectra counts of six replicates.

All statistical analyses were performed on six replicates using Scaffold software and calculation of the *P*-value using the Fisher's exact test. Scaffold computed the false discovery rate (FDR) using the Benjamini–Hochberg procedure. Selected proteins were identified with *P*-values < 0.05, FDR < 0.3% decoy, and FDR 0.1% decoy.

Finally, subcellular location of the identified proteins was annotated using the Gene Ontology Cell Component annotation tool of AMIGO 2 (http://amigo.geneontology.org/amigo/landing) using

default settings at medium stringency. The interactome of the membrane proteins with extracellularly exposed polypeptides we pulled down with Fib-Tau was obtained using the String database (String v10, https://string-db.org/; Szklarczyk *et al*, 2015) with parameters set to only detect interactions that were validated experimentally or described in databases. The resulting network was further exported to Cytoscape (version 3.5.1 at http://www.cytoscape.org/; Smoot *et al*, 2011; Saito *et al*, 2012) to visualize interactions between the identified proteins.

For the α- and β-NKA, GluA1, GluA2, GluN1, and GluN2B peptide-targeted identification strategy, database searching was performed using the Mascot search engine (Matrix Science, London, UK; version 2.4.1) against the database composed only of α- and β-NKA, GluA1, GluA2, GluN1, and GluN2B primary sequences, including cysteine carbamidomethylation as fixed modification and methionine oxidation, N-terminal acetylation, N-terminal pyroglutamylation, and monolink modification of K, S, Y, and T residues with DTSSP as variable modifications.

## Proteomics data deposition

The mass spectrometry proteomics data (neurons exposed to Fib-Tau-1N3R or 1N4R and that of control neurons) have been deposited to the ProteomeXchange Consortium (Vizcaíno *et al*, 2014) via the PRIDE partner repository. The data set identifiers are as follows: Project Accession: PXD009294, Project https://doi.org/10.6019/pxd009294 (Project name: Identification of proteins that interact with extracellularly applied fibrillar Tau assemblies).

## Primary neuronal cultures

Primary neurons were prepared from embryonic day 18 C57BL/6J mice (Janvier Labs, France). For biochemistry experiments, (proteomics studies and cell-surface biotinylation), cortical neuronal cultures were plated on 6-cm plate pre-coated with 80 mg/ml poly-D, L-ornithine (Sigma, 24–48 h; Shrivastava *et al*, 2013b, 2015). Cortical neurons were used, as they can be prepared in larger quantities ($2 \times 10^6$ cells/dish) as required for these experiments. All other experiments were performed on hippocampal neuronal cultures plated on 18 mm coverslips pre-coated (24–48 h) with 80 mg/ml poly-D, L-ornithine. Freshly dissociated (trypsin) hippocampal cells were plated ($1 \times 10^5$ cells/well) in neuronal attachment media consisting of 10% horse serum (Eurobio), 1 mM sodium pyruvate (GIBCO), 2 mM Glutamax-100X (GIBCO), and penicillin/streptomycin (GIBCO) in MEM (GIBCO) for 3 h. The attachment medium was replaced and cells were maintained in serum-free neurobasal medium (GIBCO) supplemented with 1X B27 (GIBCO) and 2 mM Glutamax-100X. Cells were maintained by supplementing with fresh serum-free neurobasal medium every week.

## Hippocampal injection and perfusion

Wild-type mice (C57BL/6J) were obtained from Janvier Labs, France, and maintained at the animal facility (École Normale Supérieure) until surgery (Shrivastava *et al*, 2015). Following anesthesia (ketamine/xylazine), animals were placed on a stereotaxis apparatus. A tiny hole in the skull was opened, and 1 μl of Fib-Tau

(3.9 μg) was pressure-injected in the CA1 region of the hippocampus at a flow rate of 150 nl/min [from the bregma: anteroposterior (AP) −2.9 mm, mediolateral (ML) +3 mm, dorsoventral (DV) −2 mm]. Mice of both gender were transcardially perfused with 4% paraformaldehyde after 8 and 24 h post-surgery. Their brains were collected, post-fixed in 4% paraformaldehyde overnight, and cryo-protected using sucrose (20%) before sectioning. 20-μm-thick coronal sections were prepared using cryostat maintained at −20°C. The sections were stored in 1× PBS (phosphate-buffered saline) for 3–4 days at 4°C until immunohistochemistry was performed.

## Thresholding, cluster identifications, and co-localization analysis

Confocal (immunohistochemistry) and spinning disk images (immunocytochemistry) were filtered by wavelet decomposition (Izeddin *et al*, 2012) using an interface that has been custom implemented in Metamorph software. Wavelet decomposition allows the separation of small and large structures (clusters) based on their fluorescence intensities and has been used in several biological studies (Bannai *et al*, 2009, 2015; Renner *et al*, 2010; Shrivastava *et al*, 2013a,b, 2015). Here, this approach was used to generate background-free masks to identify the clusters of a specific protein of interest such as Fib-Tau, homer, α3-NKA, GluA1-2-AMPA, and GluN1-2B-NMDA. "*Number of clusters/μm²*" was obtained by counting the total number of clusters within a manually drawn region of interest (ROI, both axons and dendrites). "*Intensity of cluster*" is defined as total fluorescence intensity per cluster. For each image, the values of all clusters were averaged within the ROI. "*Co-localization*" is defined when there was an overlap between the thresholded clusters of two images. Since immunocytochemistry was performed post-permeabilization, the intensity of clusters of only synaptically co-localized α3-NKA, GluA1-2-AMPA, and GluN1-2B-NMDA was quantified.

## Stochastic optical reconstruction microscopy (STORM) of TauF3R and quantification

For STORM imaging, neurons were exposed for increasing time to ATTO-647N-labeled Fib-Tau (Appendix Fig S2F) prior to paraformaldehyde (4%) fixation. Imaging was performed under reducing condition with buffer composed of PBS (pH 7.4), glucose (10%), β-mercaptoethylamine (50 mM), glucose oxidase (0.5 mg/ml), and catalase (40 mg/ml), and deoxygenized with nitrogen (Shrivastava *et al*, 2015). A total of 20,000 frames were acquired. STORM imaging was carried out on an inverted Nikon Eclipse Ti microscope equipped with a 100× oil immersion objective (N.A. 1.49 with a microscope-inbuilt 1.5× lens) using an Andor iXon EMCCD camera (image pixel size, 106 nm). ATTO-647N was imaged using laser 639 nm (1 kW, used at 500 mW) for a 50-ms exposure time. Single molecules were detected and rendered with a pointing accuracy of 10 nm (Gaussian radius, 10 nm) using MATLAB (Shrivastava *et al*, 2013b). All the quantifications were performed using open-source softwares ImageJ and Lama (Malkusch & Heilemann, 2016), the latter was used to compute DBSCAN algorithm (Ester *et al*, 1996). This approach allows identifying clusters in large spatial data sets by looking at the local density of points. Here, after correcting for multiple detections in consecutive frames,

"density threshold" of minimum 20 detections within a radial distance of 20 nm was used. The proportion of single-particle detection events within Fib-Tau clusters corresponds to the number of detections within clusters divided by the total number of detections.

### Single-particle quantum dot labeling and analysis

Quantum dot (QD)-based single-particle tracking (SPT) protocol and analysis methods have been used and described in several previous publications (Bannai *et al*, 2006; Lévi *et al*, 2008; Renner *et al*, 2010; Shrivastava *et al*, 2013a, 2015). For SPT of biotin-labeled Fib-Tau (Appendix Fig S1), first, cells were exposed to biotinylated fibrils. This was followed by labeling a small proportion of Biotin-Fib-Tau using streptavidin-QD-605 nm (1:10,000, 2–3 min). For SPT-QD experiments of AMPA and NMDA receptors, neurons were transfected with pHluorin-tagged GluA1/GluA2/GluN1/GluN2 constructs (see table above). Single pHluorin molecules were tracked using GFP antibody pre-coupled with QD-655 nm (pre-coupling protocol: 1 μl rabbit-GFP antibody + 0.5 μl Fab'-QD-655 + 7.5 μl 1× PBS, mixed and gently shaken for 30 min, then 1 μl 1× casein was added and shaken for additional 15 min). Transfection was performed on DIV 18, and experiment was performed on DIV 21. Live neurons were exposed to Fib-Tau in the culture medium and in an incubator maintained at 37°C/5% $CO_2$. Unbound Fib-Tau was washed before experiments. All the washing and imaging was performed in MEM recording medium (phenol red-free MEM, 33 mM glucose, 20 mM HEPES, 2 mM glutamine, 1 mM sodium pyruvate, and 1× B27). Neurons were imaged at 37°C using an inverted microscope (IX71, Olympus) equipped with an oil immersion objective (Olympus, 60× with 1.5×, NA1.45), a xenon lamp, and cooled CCD camera Cascade + 128 (Roper Scientific). Fluorescent signals were detected using appropriate filter sets (QD: D455/70× and HQ655/20; GFP, HQ500/20 and HQ535/30).

Tracking and analysis were performed using SPTrack_v4.2, homemade software in MATLAB (MathWorks; Renner *et al*, 2010). The center of the QD fluorescence spot was determined by Gaussian fitting with a spatial resolution of about 10–20 nm. The spots in a given frame were associated with the maximum likely trajectories estimated on previous frame of the image sequence. Trajectories with a minimum length of at least 15 consecutive frames were considered. The *Mean Square Displacement (MSD)* was calculated using:

$$MSD(ndt) = (N - n)^{-1} \sum_{i=1}^{N-n} [(x_{i+n} - x_i)^2 + (y_{i+n} - y_i)^2],$$

where $x_i$ and $y_i$ are the coordinates of an object on frame I, $N$ is the total number of steps in the trajectory, $dt$ is the time between two successive frames, and $ndt$ is the time interval over which displacement is averaged (Saxton & Jacobson, 1997). The *Diffusion Coefficient (D)* was calculated by fitting the first two to five points of the MSD plot versus time with the equation

$$MSD(t) = 4D_{2-5t} + 4\sigma_x^2,$$

with $\sigma_x$ is the spot localization accuracy in one direction. The *Explored Area* represents the distribution of MSD values for a given interval of time. In this study, the explored area of individual trajectories represents the area (μm$^2$) covered by trajectory

between 0.6 and 0.9 s. The analysis of the explored area reveals the heterogeneity of the diffusion in the population of trajectories and allows applying statistical tests on MSD data.

### Single-particle tracking—photoactivated localization microscopy (SPT-PALM) and analysis

SPT-PALM imaging was performed on DIV 21 on live hippocampal neurons, transfected on DIV 18 with α3-NKA-Dendra2 plasmid as previously (Shrivastava *et al*, 2013b). Prior to conversion, Dendra2 has excitation and emission maxima of 490 and 507 nm (green range) while following conversion Dendra2 protein has excitation and emission maxima of 553 and 573 nm (red range). First, all signal in red channel was photo-bleached to allow detection of single molecule events arising due to the photo-conversion of Dendra2 from green to red channel. Single molecule events of Dendra2 were acquired at 50 Hz (20 ms) using laser 561 nm (0.5 kW, used at 200 mW) while pulse-activating with 405 nm laser (100 mW power, used at 2–5 mW) for 6,000 frames. Live neurons were exposed to Fib-Tau in the culture medium and in an incubator maintained at 37°C/5% $CO_2$. Unbound Fib-Tau was washed before experiments. All the washing and imaging was performed in MEM recording medium (phenol red-free MEM, 33 mM glucose, 20 mM HEPES, 2 mM glutamine, 1 mM sodium pyruvate, and 1× B27).

Detection and tracking were done with a multiple target tracing (MTT)-based (Sergé *et al*, 2008) analysis software custom developed by M.R. (SuperRes v.1). The center of each fluorophore signal was determined using a Gaussian fit with a localization precision of about 20 nm. Trajectories of a minimum of five detections (six time points) were considered for analysis. MSD curve and D were calculated as described above.

#### *Hidden Markov model*
The most probable model of diffusive states was inferred by a modified version of vbSPT analysis software (Persson *et al*, 2013), which applies a Bayesian treatment of hidden Markov models. Prior values of D and dwell time were 0.1 μm$^2$/s and 50 frames (1,000 ms), respectively. The minimum length of trajectory was 2. The script was run to let freely choose the most probable model, with a maximum of three possible states. For α3-NKA-Dendra2, two possible states were detected in most cases. Based on the model, the diffusion coefficient, the proportion of cells in each state, dwell time in each state, and the transition probability (occupancy) per time step were calculated and shown.

### Statistics and graph representations

In all imaging studies, the distribution of values has been plotted either as box-plot or as scatter dot plots. For diffusion studies, cumulative distribution plot has been plotted to represent the differences as in previous studies (Lévi *et al*, 2008; Bannai *et al*, 2009; Renner *et al*, 2010; Shrivastava *et al*, 2015). An effect is considered real only if it is observed in all individual experiments. Unless otherwise specified, all the statistical analysis test on microscopy data we performed is Mann–Whitney test, which is a non-parametric test of distribution, i.e., it does not make any assumption about the distribution of the two population being compared. Graphs have been prepared using GraphPad Prism version 5, and figures have been

prepared using Adobe Illustrator CS5, ImageJ/Fiji, and Microsoft Excel 2016.

**Expanded View** for this article is available online.

## Acknowledgements

The authors thank Tracy Bellande for expert technical assistance, David Cornu for expert assistance in mass spectrometry, and Clément Lena for his support. This work was supported by Grants from the Agence Nationale de la Recherche (ANR-14-CE13-0031-02 SPREADTAU), the EC Joint Programme on Neurodegenerative Diseases (JPND-NeuTARGETs-ANR-14-JPCD-0002-02; JPND-SYNACTION-ANR-15-JPWG-0012-03), the Centre National de la Recherche Scientifique, the Institut National de la Santé et de la Recherche Médical, The Fondation pour la Recherche Médicale (Contract DEQ 20160334896), a "Coup d'Elan a la Recherche Francaise" award from Fondation Bettencourt-Schueller, the Fondation Simone et Cino Del Duca of the Institut de France, the Innovative Medicine Initiative 2 grant agreement No 116060 (IMPRiND, www.imprind.org) supported by the European Union's Horizon 2020 research and innovation program and EFPIA, an ERC advanced research grant (PLASLTINHIB), and the "Investissements d'Avenir" program (ANR-10-LABX-54 MEMO LIFE and ANR-11-IDEX-0001-02 PSL Research University). This work benefited from the electron microscopy facility Imagerie-Gif and the proteomic facility SICaPS.

## Author contributions

RM, ANS, and AT: conceived the project, designed experiments, and wrote the manuscript. ANS, VR, LP, LBo, KM, CM-H, and AC: performed experiments. ANS, VR, and RM: analyzed data. MR: developed tools for data analysis. LBu and PH: provided tools and reagents for experiments.

## Conflict of interest

The authors declare that they have no conflict of interest.

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
