## [Review Process File · The EMBO Journal]

Clustering of Tau Fibrils Impairs the Synaptic Composition of $\alpha 3$ -Na⁺/K⁺-ATPase and AMPA receptors

Amulya Nidhi Shrivastava, Virginie Redeker, Laura Pieri, Luc Bousset, Marianne Renner, Karine Madiona, Caroline Mailhes-Hamon, Audrey Coens, Luc Buée, Philippe Hantraye, Antoine Triller and Ronald Melki

Review timeline:

Submission date:	21st May 2018
Editorial Decision:	4th Jul 2018
Revision received:	28th Sep 2018
Editorial Decision:	29th Oct 2018
Revision received:	20th Nov 2018
Editorial Decision:	28th Nov 2018
Revision received:	29th Nov 2018
Accepted:	30th Nov 2018

Editor: Karin Dumstrei

Transaction Report:

1st Editorial Decision

4th Jul 2018

Thank you for submitting your manuscript to The EMBO Journal. Your study has now been seen by three referees and their comments are provided below.

The referees appreciate the analysis and insights made. However, they also raise a number of different concerns that should be resolved for consideration here. In particular there is a need for more controls and the writing style has to be improved. Should you be able to extend the findings along the lines as indicated by the referees then we would like to invite a revised version.

Given the referees' positive recommendations, I would like to invite you to submit a revised version of the manuscript, addressing the comments of all three reviewers. I should add that it is EMBO Journal policy to allow only a major single round of revision, and that it is therefore important to resolve the raised concerns at this stage.

REFeree REPORTS

Referee #1:

In their study, Melki and colleagues used fragmented, fibrillar preparations of 1N3R or 1N4R Tau

(Fib-Tau) and showed that these preparations bind to the neuronal plasma membrane and eventually form clusters at excitatory synapses. They then used pull-down approaches to identify the associated proteins by mass spectrometry. They focused on an analysis of the NKA complex $\alpha 1$, $\alpha 3$ and $\beta 1$ subunits, AMPA receptor GluA1 and GluA2 subunits and NMDA receptor GluN1 and GluN2B subunits. In particular they showed that the clustering results in a reduction in NKA and an increase in the AMPAR subunit GluA2 at synapses.

More generally, I think this is an interesting approach using a host of in vitro and in vivo techniques including super resolution microscopy and mass spectrometry. It is interesting to observe that rather than focusing on the toxicity of oligomers, this manuscript focuses on fibrils that have regained interest in the wake of the prion hypothesis of tauopathies.

In general terms, slight language editing (including syntax) could help improving this manuscript. I also feel that some details are buried in the Materials and Methods section and should be added to the main text, such as the fact that the fibrils which are being added are fragmented (only mentioned in legend Fig S1 as far as I can tell). Also, while referencing previous work on alpha-synuclein is fine, the current manuscript should be a stand-alone document not requiring to read the earlier paper.

My specific comments are as follows:

My major concern is around the specificity of the clustering on excitatory synapses (see e.g. comments 7 and 13).

Introduction:

- 1 - FTDP-17 is not a form of PD - it is a genetically inherited form of tauopathy.
- 2 - The concept of 'naive neighbor cells' taking up and propagating seeds is too simplistic. Uptake is considered to occur along projections and toxicity often spares neighbouring neurons or other cells such as glia.
- 3 - We assess the 'very first and key step' - this should be removed as it is unclear what the first and key step is.
- 4 - Whether tau stabilises microtubules is not entirely clear - tau is clearly associated with microtubules.

Results (plus figures):

- 5 - Figure 1A: What is the rationale for using 3 males for the 8h time-point and 3 females for the 24h time-point? Fig 1B show needle track. Why does one not see the lateral diffusion pattern in the CA region shown in the following panels?
- 6 - page 5: How is a cluster defined? Why would cluster formation reduce total fluorescence?
- 7 - page 6, Fig 1: The authors claim to see clustering on the plasma membrane of primary neurons. When one examines Supplementary Fig 4, it looks more like random precipitates forming, some sedimenting on the cell and its processes and others on the surface of the tissue culture plate. I would therefore suggest to add a control plate with no cells at all. The fact that 'diffusion is confined after 60 min' may simply be related to the fact that by then (compared to 10 min) the surface is more crowded. Reference to fig S2C again raises the issue of sedimentation rather than clustering. The last sentence on this page should be revisited. Continued on top of page 7 the claim is made that 'most, if not all Fib-Tau clusters are at the cell surface upon exposure for 1h (Figure 2G, overlay).' This is contradicted by Fig S4.
- 8 - Page 7 2nd paragraph should be reworded. 'Diffusion-dependent clustering of exogenous Fib-Tau' cannot be claimed at this stage. The 'time-dependent increase in cluster size' (Fig 2B) requires to show that the fragmented fibrils on their own, without a cellular interaction, don't form larger clusters (also, aggregates may be the better term).
- 9 - Page 8 '50% of Fib-tau reside within the clusters at any given time-point...' It is not clear to me how this has been concluded?
- 10 -Page 4: While so far the data are shown for hippocampal cultures, this analysis switches to cortical cultures. What is the rationale for this change?
- 11 - Fragmented tau fibrils are an 'artificial' preparations (although size fragmentation admittedly allows for some control). For the mass spec analysis I am wondering whether it would be good (would have been good) to have a control preparation of either monomeric or oligomeric tau or an unrelated fibrillar entity. However, without repeating the mass spec analysis this could be easily done for the co-IP experiments (Fig 4E).

- 12 - Figure 4 mass spec: How many membrane proteins with no extracellular domain were identified as 'interacting'?
- 13 - Page 10 Fig 5: Claim of specificity of Fib-tau for excitatory synapses: the authors claim Fib-Tau colocalization for 10-20% of the excitatory synapses. They say that inhibitory synapses comprise 10-15% of all synapses which means that one would expect roughly 1% co-localization if the frequency were the same. This is what we see and yet, the authors claim specificity.
- 14 - In vivo co-localization studies: How thick were the sections? A good method would be array tomography to show co-localization.
- 15 - Page 12: Does the size of the GluA clusters increase (as claimed) after Tau-Fib treatment?
- 16 - Page 13 Super res: How do the authors account for spines being perpendicular (i.e. when no discrimination is possible between spine and shaft)? I don't understand the conclusion end of 2nd paragraph (nor the first sentence).
- 17 - Page 14 Synuclein experiment: This experiment should be repeated with plates onto which no cells have been seeded. On the other hand, I do think that this part is incomplete and I would almost be inclined to suggest removing it from the manuscript.

Discussion

- 18 - page 17 comment on prion-like propagation (twice) could be removed.

Referee #2:

Summary

This paper investigates the binding of exogenous 3R and 4R Tau assemblies (Fib-Tau) to the plasma membrane of mouse neurons in cell culture. The authors performed a proteomic screen using pull-down experiments to identify proteins of the neuronal plasma membrane that interact with exogenous applied Fib-Tau. They confirmed some of the hits with additional crosslinking/ LC-MS and co-immunoprecipitation and identified the alpha3 NKA subunit, the GluA2-AMPA and the GluN1-NMDA as interactors with Fib-Tau. They subsequently demonstrate that Fib-Tau co-localizes with excitatory synapses and the identified cell membrane receptors. They employ live-neuron SPT-QD and SPT-PALM to show that the presence of Fib-Tau can affect the dynamics of the identified receptors at the plasma membrane. Finally, they show that the presence of alpha-synuclein fibrils enhances the clustering of Fib-Tau at the plasma membrane.

Strengths of the paper are that it is well written and clearly organized. The authors give a very detailed and thorough description of the experimental techniques employed in this study. The observations on co-localization and changes of membrane dynamics of several receptors upon treatment with Tau-Fib are interesting and novel. However, the functional relevance of the observations for the prion-like spread of tau remains unclear, and very little mechanistic insight is provided throughout the paper. In addition, the paper lacks important controls to validate the findings, e.g. knockout experiments of the identified receptors to validate the functional relevance, and a comparison of the changes of membrane dynamics to negative control proteins. The observation that tau clustering increases in the presence of alpha-synuclein is interesting but remains purely observational. Thus, the current version of this work would need major revisions for a publication in EMBO.

General comments:

- The interpretation of the image data (e.g. figure 2A, G and H; figure 3, figure 5 etc) is difficult for the reader since the magnification is low, and there is no nuclear stain (or other reference stain) that facilitates the orientation within the image. If possible, please include inlays with higher magnification and/or images with additional stains to facilitate data interpretation.
- The data would be more convincing if the authors could show that the binding of Fib-Tau to glutamate receptors and NKA has a functional relevance (e.g. increased cellular tau uptake or intracellular seeding, and/or significant increase of calcium signaling or toxicity). Also, it would enhance the impact of the paper if the authors provided evidence that they can block Fib-Tau co-localization with the receptors by pharmacological inhibition and/or genetic manipulation (e.g. knockdown of receptor or receptor subunits).
- The experiments assessing membrane dynamics with SPT-QD and SPT-PALM should be

performed in comparison to an appropriate negative control protein, to show that the observed effects are specific to tau.

Specific comments

p. 5 and figure 1:

- The injection of tau fibrils into mouse brain has been extensively studied by other authors (Guo et al., 2016, Kaufman et al., 2016 and others, as cited by the authors). It is not clear how this experiment fits into the narrative of the paper, since all the subsequent experiments were done in cell culture. This data should be placed in the supplement.
- P. 5, 1st paragraph: Why did the authors choose male animals for the 8 hrs time point and female animals for the 24 hrs time point? It would be more informative to choose the same number of animals from each gender for each time point. Please comment.
- How do the authors explain the rapid reduction of tau after 24 hrs? It would have been interesting to include a fluorescently labeled control protein, to see if the rapid clearance occurs in a similar way. Additional immunohistochemistry for tau aggregates could rule out that that rapid tau clearance is caused by quenching of ATTO-550 and enhance the results of this experiment.

p. 6 and Figure 2: The authors show in figure 2G that most tau aggregates co-localize with the plasma membrane after an exposure of 60 min. However, this might not apply for figure 2A, since more aggregates might be localized intracellularly with increasing exposure time. This aspect should be discussed in the legend or in the text. Ideally, the co-localization experiment should be done for different time points to assess uptake over time.

p. 7 and figure 2H: The authors observe that only a few tau clusters remain on the cell membrane after washing and recovery of the cells for 60 min. They conclude that the tau fibrils either dissociate from the cells, or are taken up into the cells. However, they do not provide evidence for uptake versus dissociation to support this assumption. The fate of the fibrils after binding seems to be essential from a functional standpoint, and should be assessed by additional experiments, e.g. quantification of intracellular tau fibrils (labeled with a fluorescent dye).

p. 10 and figure 4F: It would make it easier for the reader if the authors could provide a figure that compares the major binding partners for 3R tau and 4R tau, and to what degree they overlap.

p. 11: It would be helpful for the reader if the authors could include a schematic in the supplement, showing the polypeptide sequence of GluA2 and the localization of the crosslink with Fib-Tau. Also, were the authors able to derive any information on the tau residues that are involved in this crosslink? Please comment.

Suppl. Figure 9: The increase of the calcium response upon exposure with tau is very subtle. What are the conclusions about the functional relevance of tau binding to the receptors, based on this experiment and the relatively limited impact on calcium signaling? The experiment should be repeated with higher tau concentrations, to see if this will increase the observed effects on calcium signaling.

Figure 7 and p. 12-14: It would increase the quality of the findings demonstrated with SPT-PALM if the authors could provide data for a control protein that does not interact with the receptors investigated here and thus does not induce the observed effects (e.g. reduction of the proportion of alpha3-NKA molecules in the Bound State upon exposure to Tau-Fib)

p. 14, top of page, and figure 7J: The authors imply that Tau-Fib uptake might be mediated by clathrin-mediated endocytosis, since there is co-localization of Tau-Fib and clathrin pits in Figure 7J. It would enhance the impact of this observation if the authors could provide data for a functional relationship between NKA, clathrin, and Fib-Tau (e.g. by genetic manipulation of clathrin-mediated endocytosis, and/or the use of pharmacological inhibitors).

p. 16, 2nd paragraph: The authors discuss that it might be "likely that the displaced NKA alpha3 subunit is endocytosed together with Fib-Tau clusters by neurons". The paper does not provide scientific evidence that this might be the case. Please change this statement accordingly or explain in more detail how this conclusion was made.

Referee #3:

Accumulation of the tau protein is the defining pathological feature of a class of neurodegenerative diseases collectively referred to as tauopathies, that include Alzheimer's disease. A wealth of recent evidence suggests that tau oligomers can propagate in a prion-like manner. In this manuscript, Shrivastava et al. explore the binding of tau fibrils to neurons. Importantly, they found that tau fibrils interacted with the Na⁺/K⁺-ATPase and AMPA receptors. They also show that tau clustering leads to a redistribution of the $\alpha 3$ subunit of Na⁺/K⁺-ATPase and of the GluA2 subunit of the AMPA receptor. Finally, they show that α -synuclein enhances tau clustering. While potentially interesting, the pathophysiological relevance of the data presented needs to be clarified.

Major points

- The Results section is not easy to read. It contains a large amount of practical information that should be restricted to the Materials and Methods section. The Results section should be extensively edited. The same comment also applies to some of the legends.
- Figures 1C and 1D must be shown with the same image settings.
- While investigating time-dependence of tau clustering makes sense (Fig. 2A), as it is a dynamic process, what is the pathological significance of investigating concentration-dependence? What would a physiologically meaningful concentration be? The same comment applies to α -synuclein in Figure 8.
- It would be important to compare proteins binding to monomeric tau with those binding to fibrillar tau. Was this done?
- The immunoprecipitation validation of mass spectrometry data (Fig. 4E) is obviously essential. This must be done by Western blotting, a slot blot is not appropriate. Since detection of biotinylated fibrillar tau was done with streptavidin-HRP, how would IgGs interfere with detection? This should also be done for 1N3Rtau fibrils as well as 1N4Rtau fibrils.
- Why are data on 1N3Rtau fibrils presented in the main section of the manuscript and the data on 1N4Rtau as supplementary data rather than the other way round? This would be more logical as, although both isoforms are expressed in the adult human brain, 1N4Rtau is adult-specific.
- The authors should discuss how heparin-induced tau filaments represent a good model for pathological tau species.

Minor points

- The end of the Introduction is far too long. As it stands, it looks like an extended abstract of the whole manuscript. It should be limited to a few sentences stating the main results and their significance.
- Legends of figures from Figure 2 should indicate that the experiments have been performed in cultured neurons.
- 'string' should be written 'String' when referring to the database.
- Page 14. 'labs' should be 'laboratories'.

1st Revision - authors' response

28th Sep 2018

Referee #1:

In their study, Melki and colleagues used fragmented, fibrillar preparations of 1N3R or 1N4R Tau (Fib-Tau) and showed that these preparations bind to the neuronal plasma membrane and eventually form clusters at excitatory synapses. They then used pull-down approaches to identify the associated proteins by mass spectrometry. They focused on an analysis of the NKA complex $\alpha 1$, $\alpha 3$ and $\beta 1$ subunits, AMPA receptor GluA1 and GluA2 subunits and NMDA receptor GluN1 and GluN2B subunits. In particular they showed that the clustering results in a reduction in NKA and an increase in the AMPAR subunit GluA2 at synapses.

More generally, I think this is an interesting approach using a host of in vitro and in vivo techniques including super resolution microscopy and mass spectrometry. It is interesting to observe that rather than focusing on the toxicity of oligomers, this manuscript focuses on fibrils that have regained interest in the wake of the prion hypothesis of tauopathies.

In general terms, slight language editing (including syntax) could help improving this manuscript. I also feel that some details are buried in the Materials and Methods section and should be added to the main text, such as the fact that the fibrils which are being added are fragmented (only mentioned in legend Fig S1 as far as I can tell). Also, while referencing previous work on alpha-synuclein is fine, the current manuscript should be a stand-alone document not requiring to read the earlier paper.

We thank reviewer 1 for her/his comments. The changes we made within the manuscript are highlighted in yellow. We have carefully read and improved the text and added necessary methodological details, wherever necessary in a context where another reviewer is suggesting we move all methods to the supplementary material. We mention that fibrils are fragmented (Page 4) and describe how we prepared them. The fragmentation protocol was clearly described in Material and Methods section (Appendix Text Page 13).

While we agree that the manuscript should stand by itself and believe it does, we very well understand the binding of Fib- α -Syn to membranes and believe it is relevant to compare the behaviour of Fib-Tau to Fib- α -Syn. The two kinds of fibrils share a common binding partner (α 3-NKA) with an opposite effect on α 3-NKA membrane diffusion/clustering. In addition, we provide evidences that Fib- α -Syn can modify Fib-Tau binding, but not that of Fib-A β in the revised version of the manuscript.

My specific comments are as follows:

My major concern is around the specificity of the clustering on excitatory synapses (see e.g. comments 7 and 13).

Introduction:

1 - FTDP-17 is not a form of PD - it is a genetically inherited form of tauopathy.

We have corrected the sentence (Page 3).

2 - The concept of 'naive neighbor cells' taking up and propagating seeds is too simplistic. Uptake is considered to occur along projections and toxicity often spares neighbouring neurons or other cells such as glia.

We have removed the word naive. Even though pathogenic Tau release sites is still unclear, we changed the sentence to "They are released by projections of affected neuronal cells, are taken up by unaffected cells and amplify by seeding the aggregation of endogenous Tau" (Page 3). We would like nonetheless to stress that Tau assemblies are found in the extracellular medium and CSF. Thus, it is still early to rule out that Tau does not bind to non-connected neurons and non-synaptic regions close to their release site.

3 - We assess the 'very first and key step' - this should be removed as it is unclear what the first and key step is.

We deleted this sentence.

4 - Whether tau stabilises microtubules is not entirely clear - tau is clearly associated with microtubules.

Most of the studies in the last 3-decades point to stabilization of microtubules by Tau. We do not intend to challenge those studies even if a recent study suggest otherwise (Qiang et al., Tau Does Not Stabilize Axonal Microtubules but Rather Enables Them to Have Long Labile Domains. Curr Biol. 2018 Jul 9;28(13):2181-2189.e4)

Results (plus figures):

5 - Figure 1A: What is the rationale for using 3 males for the 8h time-point and 3 females for the 24h time-point? Fig 1B show needle track. Why does one not see the lateral diffusion pattern in the CA region shown in the following panels?

We apologize for not being sufficiently clear. Several such injections were performed. We did not observe gender-dependent differences in exogenous Fib-Tau diffusion within the time frame 6-24h. We chose to present in this study observations made at time points 8h and 24h and animals of the

same gender for each point. As exogenous Fib-Tau diffusion was observed at both time points, we quantified both these data sets.

Figure 1B does not show the needle track. It shows the corpus callosum where most of injected Fib-Tau was detected in all 6-animals as described Page 5. The injection was performed in the CA1 cell-body layer (different z-plane). Since the fluorescence level is so high in the corpus callosum, the images shown in Fig 1B were acquired using very low exposure time. Therefore, the diffusion in the various region is not visible in this specific image. The distribution in other layers are shown Fig 1C, 1D and Appendix Fig 3.

6 - page 5: How is a cluster defined? Why would cluster formation reduce total fluorescence?

We have a separate section in the methods section “Thresholding, Cluster identifications, and Co-localization Analysis” that specifically address this issue (Page 30).

For the second question, we believe the reviewer refers to Figure 1G-H and 2I-J. Here total fluorescence is reduced concomitantly with the decrease in the number of clusters between 8h and 24h.

7 - page 6, Fig 1: The authors claim to see clustering on the plasma membrane of primary neurons. When one examines Supplementary Fig 4, it looks more like random precipitates forming, some sedimenting on the cell and its processes and others on the surface of the tissue culture plate. I would therefore suggest to add a control plate with no cells at all. The fact that 'diffusion is confined after 60 min' may simply be related to the fact that by then (compared to 10 min) the surface is more crowded. Reference to fig S2C again raises the issue of sedimentation rather than clustering. The last sentence on this page should be revisited. Continued on top of page 7 the claim is made that 'most, if not all Fib-Tau clusters are at the cell surface upon exposure for 1h (Figure 2G, overlay).' This is contradicted by Fig S4.

The reviewer got a wrong impression. There are no random precipitates formed in Appendix Figure S4. All the images are acquired several μm above the plane of the coverslip. Appendix Figure S4 (updated version below, **Fig A**) shows a double immunolabeling of axons (green) and dendrites (blue). Axons cover much larger surface in cultured neurons as evident below where we see a large proportion of Fib-Tau (red) binding. In addition, we plated cells at a much higher density (100,000 cells per 18mm coverslip). The purpose of this figure is to show that Fib-Tau can interact with the cellular processes and not only with axon/dendrite as indicated on page 5. This is further supported by identification of proteins that are enriched in axons (Neurexins), dendrites (Glutamate receptors) or both ($\alpha 3$ -NKA).

Fig A. Binding of Fib-Tau to cultured neurons (Appendix Figure S4)

The claim that Fib-Tau predominantly cluster on cell surface is based on Figure 2G. Here neurons were exposed to Fib-Tau that were labelled with both biotin and ATTO488 dye. Fib-Tau on cell surface was labelled by Streptavidin-550. As the vast majority of the clusters are ATTO488 and Streptavidin 550 positive, we conclude that “*most, if not all Fib-Tau clusters are at the cell surface upon exposure for 1h*” (Page 6). Following reviewer #2 comments, we provide an additional Figure (Appendix Figure S5) that shows that clusters of Fib-Tau are primarily localized on the membrane at

least until 4-hours. In this manuscript, we are not studying Fib-Tau endocytosis or the state of Fib-Tau during endocytosis (single molecule/clustered). This issue has been partially dealt with in Flavin et al., 2017 *Acta Neuropathol* 134:629-653.

The reviewer states that “The fact that 'diffusion is confined after 60 min may simply be related to the fact that by then (compared to 10 min) the surface is more crowded’”. This is indeed the case as described in our review article (Shrivastava et al, 2017, *Neuron*). Plasma membrane favours clustering of Fib-Tau due to inter-molecular interactions in a crowded environment. The higher the concentration of Fib-Tau, the more clusters will be detected, which is what we report here. The mean diffusion coefficient values are $0.085 \mu\text{m}^2/\text{s}$ (unexposed) and $0.055 \mu\text{m}^2/\text{s}$ (Fib-Tau), consistent with the mobility of proteins. Immobile molecules or Quantum dots sticking on the coverslip will typically exhibit diffusion coefficient in order of $10^{-4} \mu\text{m}^2/\text{s}$. Note that we do not see a dramatic slow-down in Fib-Tau diffusion (Figure 2E). This suggests that Fib-Tau single particles are slowed-down/clustered but are not immobile on the membrane. These clusters then act as deleterious scaffolds for membrane proteins.

Lastly, coverslips have charged surfaces (typically hydrophilic) that may bind charged fibrils but also proteins present in the culture medium through electrostatic interaction. To further demonstrate preferential binding of Fib-Tau to cells under our experimental conditions, we exposed non-neuronal HEK cells (**Fig B**) grown on the same coverslip used in neuronal cultures. As the reviewer can see from the figure below, most of Fib-Tau is bound to cells or is intracellular following exposure of cells for 10 and 60 min.

Figure B. Binding of Fib-Tau (0.36nM, 10min) to HEK cells

8 - Page 7 2nd paragraph should be reworded. 'Diffusion-dependent clustering of exogenous Fib-Tau' cannot be claimed at this stage. The 'time-dependent increase in cluster size' (Fig 2B) requires to show that the fragmented fibrils on their own, without a cellular interaction, don't form larger clusters (also, aggregates may be the better term).

We respectfully disagree with the reviewer. We do show binding to cells as explained in point 7. We prefer to use the term clusters instead of aggregates since we have reported using kinetic measurements the same behaviour for other pathogenic assemblies namely $A\beta$ and α -Syn (Shrivastava et al., 2015. *EMBO J*, 34, 2408-2423, Renner, et al., 2010. *Neuron* 66, 739–754, and Shrivastava, et al. 2013. *Glia* 61, 1673–1686.).

9 - Page 8 '50% of Fib-tau reside within the clusters at any given time-point...' It is not clear to me how this has been concluded?

We apologize for not being sufficiently clear. The proportion of single particles detection events within Fib-Tau clusters represents number of detections within clusters divided by total number of detections. This information has now been added in the Methods section (Page 19). In all cases, this information was computed and plotted as shown in Figure 3F.

10 -Page 4: While so far the data are shown for hippocampal cultures, this analysis switches to cortical cultures. What is the rationale for this change?

For biochemistry experiments, we needed large quantities of cells/proteins. This is why cortical neurons were used. Note that both hippocampal and cortical cultures have predominantly excitatory neurons. We would like to stress that this work is intended to identify neuronal proteins that interact with Fib-Tau not proteins potentially specific to different cell types such as hippocampal or cortical neurons.

11 - Fragmented tau fibrils are an 'artificial' preparations (although size fragmentation admittedly allows for some control). For the mass spec analysis I am wondering whether it would be good (would have been good) to have a control preparation of either monomeric or oligomeric tau or an unrelated fibrillar entity. However, without repeating the mass spec analysis this could be easily done for the co-IP experiments (Fig 4E).

Tau oligomers and monomers are certainly of interest and it is important to identify their partners. In our hands, they do not exhibit prion-like propensity. Nonetheless, we did assess the binding of monomeric Tau1N3R and 4R to neurons. These data were not shown. We provide these data in the revised version of our manuscript (Figure EV4). We did observe clusters of monomeric Tau; however, these clusters were clearly inside the neurons upon exposure for as little as one hour (Figure EV4A and EV4E). This is why, we believe they are not suitable controls as their localization and the effects they may exert on their partner proteins is different from that of fibrillar Tau. Nonetheless, we now performed SPT-PALM on neurons exposed to monomeric Tau. This technique is more sensitive to detect interactions (Figure EV4B-D) than co-IP. We found no change in the diffusion behaviour of α 3-NKA following exposure to monomeric Tau.

12 - Figure 4 mass spec: How many membrane proteins with no extracellular domain were identified as 'interacting'?

Figure 4C shows the distribution of 372 synaptic and membrane protein interactors of Fib-Tau 1N3R identified in pull down experiments. As indicated, the GO cell component annotation tool of AMIGO 2, annotated a total of 251 membrane proteins corresponding to 13 pre-synaptic membrane proteins, 34 post-synaptic membrane proteins and 204 other membrane proteins. 29 out of these 251 membrane proteins have extracellular domains. The remaining 222 proteins have no extracellular domain.

13 - Page 10 Fig 5: Claim of specificity of Fib-tau for excitatory synapses: the authors claim Fib-Tau colocalization for 10-20% of the excitatory synapses. They say that inhibitory synapses comprise 10-15% of all synapses which means that one would expect roughly 1% colocalization if the frequency were the same. This is what we see and yet, the authors claim specificity.

Our claim for specificity for excitatory synapse is not solely based on co-localization data. It is based on the fact that our proteomics study identified several scaffolding proteins enriched at excitatory synapses. This includes PSD95 (dlg4), Shank subunits, CamK-subunits, Homer, etc (Figure 4F).

The 1% co-localization with gephyrin can be specific or by chance. To determine whether Fib-Tau can interact with inhibitory synapses, we used our well-established inhibitory neuronal spinal cord cultures (Appendix Figure S7, below Fig C). Overall, exposure of spinal cord neurons to Fib-Tau yielded observations that are different from those we report for hippocampal neurons. 1) An increase in the size (fluorescence intensity) of Fib-Tau clusters was observed between 10 and 60min in spinal cord neurons that was not prominent in hippocampal neurons. 2) Fib-Tau clusters appeared completely excluded from both excitatory and inhibitory synapses. Thus, Fib-Tau binding to excitatory synapses is specific to hippocampal neurons, not spinal cord neurons. We have incorporated this set of data Page 10.

Figure C. Fib-Tau properties in spinal cord neurons (Appendix Figure S7)

14 - In vivo co-localization studies: How thick were the sections? A good method would be array tomography to show co-localization.

20 μm thick coronal sections were prepared using cryostat. We agree with the reviewer that a more sophisticated approach may have showed *in vivo* co-localization, however at this stage, we do not think it will add much additional information/weight to the manuscript/findings.

15 - Page 12: Does the size of the GluA clusters increase (as claimed) after Tau-Fib treatment?

Yes, the size of GluA2-subunit clusters increased but not that of GluA1-subunit (Figure 5D).

16 - Page 13 Super res: How do the authors account for spines being perpendicular (i.e. when no discrimination is possible between spine and shaft)? I don't understand the conclusion end of 2nd paragraph (nor the first sentence).

We only quantified the spines that could be clearly distinguished. Any regions/spines perpendicular to the shaft were not included in quantification (bright spots).

17 - Page 14 Synuclein experiment: This experiment should be repeated with plates onto which no cells have been seeded. On the other hand, I do think that this part is incomplete and I would almost be inclined to suggest removing it from the manuscript.

We have expanded this figure and show that Fib-Tau and Fib- α -Syn co-exposure increases the localization of Fib-Tau clusters specifically at synapses and over α 3-NKA which is localized ubiquitously. We also show that Fib- α -Syn effect is specific to Fib-Tau and not to monomeric-Tau (Figure EV 4E-H). In this figure, we also show that Fib-A β does not affect Fib-Tau clustering. Thus, we demonstrate that the mechanism is specific. The effect of one assembly type on Fib-Tau clustering and synaptic localization goes very well with the manuscript. We do not believe that an experiment on a coverslip will yield additional information.

Discussion

18 - page 17 comment on prion-like propagation (twice) could be removed.

The last sentence has been modified as per the suggestion.

Referee #2:

Summary

This paper investigates the binding of exogenous 3R and 4R Tau assemblies (Fib-Tau) to the plasma membrane of mouse neurons in cell culture. The authors performed a proteomic screen using pull-down experiments to identify proteins of the neuronal plasma membrane that interact with exogenous applied Fib-Tau. They confirmed some of the hits with additional crosslinking/ LC-MS and co-immunoprecipitation and identified the alpha3 NKA subunit, the GluA2-AMPA and the GluN1-NMDA as interactors with Fib-Tau. They subsequently demonstrate that Fib-Tau co-localizes with excitatory synapses and the identified cell membrane receptors. They employ live-neuron SPT-QD and SPT-PALM to show that the presence of Fib-Tau can affect the dynamics of the identified receptors at the plasma membrane. Finally, they show that the presence of alpha-synuclein fibrils enhances the clustering of Fib-Tau at the plasma membrane.

Strengths of the paper are that it is well written and clearly organized. The authors give a very detailed and thorough description of the experimental techniques employed in this study. The observations on co-localization and changes of membrane dynamics of several receptors upon treatment with Tau-Fib are interesting and novel. However, the functional relevance of the observations for the prion-like spread of tau remains unclear, and very little mechanistic insight is provided throughout the paper. In addition, the paper lacks important controls to validate the findings, e.g. knockout experiments of the identified receptors to validate the functional relevance, and a comparison of the changes of membrane dynamics to negative control proteins. The observation that tau clustering increases in the presence of alpha-

synuclein is interesting but remains purely observational. Thus, the current version of this work would need major revisions for a publication in EMBO.

We thank reviewer 2 for her/his assessment of the work. We address hereafter the issues the reviewer raises in the his/her general comment above.

The functional relevance of the observations for the prion-like spread of tau remains unclear, and very little mechanistic insight is provided throughout the paper: We do not address in this study the prion-like spreading of Fib-Tau. This is a very broad and complex issue that is not addressed in any paper to our knowledge. The manuscript is focused on one important step in this process, namely, the binding of pathogenic Tau assemblies to the plasma membrane and the immediate mechanistic consequences of this event. Very important issues are described/documentated i) we describe a diffusion-dependent binding and clustering of Fib-Tau on the membrane ii) we reveal rapid-clearance of Fib-Tau following clustering iii) we identify membrane partners using unbiased high-throughput proteomics for two-different Tau isoforms, 3R and 4R, iv) we show that Fib-Tau clusters trap GluA2-AMPA receptors at synapses leading to the redistribution of this important receptor v) we demonstrate the destabilization and increased endocytosis of $\alpha 3$ -NKA, vii) we measure increased action-potential-induced Ca^{2+} influx and reduced Na^{+} -efflux (new data), finally, viii) we demonstrate that the presence of Fib- α -Syn enhances Fib-Tau clustering, specifically at synapses (new data). This study is novel and according to us provides a new mechanistic understanding of Fib-Tau binding to neuron membranes and its consequences.

In addition, the paper lacks important controls to validate the findings, e.g. knockout experiments of the identified receptors to validate the functional relevance, and a comparison of the changes of membrane dynamics to negative control proteins: We do not understand the rationale of knocking out a single protein when Fib-Tau has at least 29-membrane partners, 5 of which we assess. Removal of a single protein at any time will not affect Tau-fibril binding to the membrane as none of these proteins can be considered as a receptor of Fib-Tau. We never conveyed the idea that Fib-Tau bind to one single protein/receptor. On the contrary, the take home message that is put forward is that Fib-Tau bind to numerous membrane proteins. These multiple interactions lead to efficient binding through numerous interactions/binders. This issue is highlighted in the discussion (Page 17) as we write “*Interfering with binding will require either the design of ligands that change the surface properties of fibrils (Melki, 2017) or shielding not one but multiple partner proteins. Thus, luring pathogenic fibrils to avoid the redistribution of membrane protein partners will require not one but multiple peptide decoys, reproducing the protein partners interaction sites.*” and is discussed in our review article (Shrivastava et al., 2017, Neuron). Fib-Tau also binds to non-neuronal cells as shown in **Figure B** (above). The dynamic measurements (**Figure 6, 7**) and localization studies (**Figure 5**) we performed with state-of-the-art methods reveal a redistribution GluA2-AMPA and $\alpha 3$ -NKA at neuronal membranes but no redistribution of GluA1-AMPA or GluN2B-NMDA. This is a compelling result that represent an internal control shall such a control be needed. We would like further to stress that knocking out $\alpha 3$ -NKA is lethal to mice. Thus, technically unfeasible. If the reviewer is suggesting to perform knockout studies on cell lines, we would face other issues such as the contribution of membrane proteins specific to those immortal cell lines to Fib-Tau binding and the relevance of such work for neurons.

The observation that tau clustering increases in the presence of alpha-synuclein is interesting but remains purely observational: This is a specific phenomenon that we observed. We found a condition under which Fib-Tau clustering is modified and their localization enhanced at synapses. We believe this observation is important and relevant for further studies. Furthermore, we have expanded this section in the revised version of the manuscript (details provided below).

General comments:

- **The interpretation of the image data (e.g. figure 2A, G and H; figure 3, figure 5 etc) is difficult for the reader since the magnification is low, and there is no nuclear stain (or other reference stain) that facilitates the orientation within the image. If possible, please include inlays with higher magnification and/or images with additional stains to facilitate data interpretation.**

All the images mentioned here were acquired at a 100X resolution using a spinning disk confocal microscope, this is how we identify and quantify individual clusters. These are diffraction-limited high-resolution (200-250nm) images. Almost all the images show section of neuronal processes

which are $0.5 \times 10\mu\text{m}$ thick. This can be seen in Figure 5, Appendix Figure 4, 5, and Figure EV2. We will not be able to depict all the information shown in Figure 2 in a single figure if we show larger imaging fields. While we understand the reviewer concerns, we did not perform nuclear staining because: i) the nuclei are in a different z-plane than plasma membrane, and ii) nuclear labelling is of no help in image interpretation when images are acquired along cellular processes.

• The data would be more convincing if the authors could show that the binding of Fib-Tau to glutamate receptors and NKA has a functional relevance (e.g. increased cellular tau uptake or intracellular seeding, and/or significant increase of calcium signaling or toxicity). Also, it would enhance the impact of the paper if the authors provided evidence that they can block Fib-Tau co-localization with the receptors by pharmacological inhibition and/or genetic manipulation (e.g. knockdown of receptor or receptor subunits).

This study is not aimed at documenting the endocytosis/uptake, seeding or toxicity of Fib-Tau. Several excellent studies dealing with such late events have been published. We already showed that 4-AP mediated Ca^{2+} -influx, that requires glutamate receptors was slightly enhanced (Figure EV3 in the revised manuscript). We added supplementary data showing that NKA-specific function is affected. Indeed, Na^+ -efflux is reduced in neurons exposed to Fib-Tau (Figure EV5, corresponding text on Page 14).

Since Fib-Tau clusters at excitatory synapses, where AMPA/NMDA receptors are localized, pharmacological and/or genetic inhibition will unlikely affect their localization and thus the early events we assess in the present manuscript such as binding. They may however have impact on late stage events.

We would like in addition to stress that we do not convey the idea that Fib-Tau bind to one single protein/receptor. We put forward that Fib-Tau bind to numerous membrane proteins. These multiple interactions lead to efficient binding through numerous interactions/binders. Thus, “blocking Fib-Tau co-localization with the receptors by pharmacological inhibition and/or genetic manipulation (e.g. knockdown of receptor or receptor subunits) is technically as suggested by the reviewer either not feasible or not particularly relevant.

• The experiments assessing membrane dynamics with SPT-QD and SPT-PALM should be performed in comparison to an appropriate negative control protein, to show that the observed effects are specific to tau.

As stated above that SPT-QD experiments were internally controlled and performed for 4-different subunits. Only one (GluA2) exhibited changes. A control for $\alpha 3$ -NKA SPT-PALM was previously lacking. We have performed additional measurements showing that the diffusion of $\alpha 3$ -NKA is not modified in the presence of monomeric Tau (Figure EV4, corresponding text on Page 13). This is most probably due to the rapid up take of monomeric Tau by neurons.

Specific comments

p. 5 and figure 1:

• The injection of tau fibrils into mouse brain has been extensively studied by other authors (Guo et al., 2016, Kaufman et al., 2016 and others, as cited by the authors). It is not clear how this experiment fits into the narrative of the paper, since all the subsequent experiments were done in cell culture. This data should be placed in the supplement.

All the studies looked at the seeding and prion-like propagation of Fib-Tau. We assess here a different issue, namely binding to neural cells. We report clustering and clearance within few hours after injection (Figure 1) that we studied mechanistically (Figure 2). We believe these are important findings that justify maintaining these data as main figures.

• P. 5, 1st paragraph: Why did the authors choose male animals for the 8 hrs time point and female animals for the 24 hrs time point? It would be more informative to choose the same number of animals from each gender for each time point. Please comment.

Several such injections were performed. We did not observe gender-dependent differences in exogenous Fib-Tau diffusion within the time frame 6-24h. We chose to present in this paper observations made at time-points 8h and 24h and animals of the same gender for each point. As exogenous Fib-Tau diffusion was observed at both time points, we quantified both these data sets.

• How do the authors explain the rapid reduction of tau after 24 hrs? It would have been interesting to include a fluorescently labeled control protein, to see if the rapid clearance occurs in a similar way. Additional immunohistochemistry for tau aggregates could rule out that that rapid tau clearance is caused by quenching of ATTO-550 and enhance the results of this experiment.

The reduction of Fib-Tau after 24h observed in Stratum oriens is due to cellular uptake as can be seen in Figure 1C-D. In addition, most of the Fib-Tau we injected in the CA-cell body were up taken in the corpus callosum (both at 8h and 24h). Therefore, the clearance is not caused by quenching, but due to uptake. In our past study (Shrivastava et al., 2015, EMBOJ), we injected ATTO550-labeled Oligomeric/Fib- α -Syn and looked at their distribution after 8h and 24h. We did not observe rapid clearance of Fib- α -Syn. Therefore, the rapid clearance seems to be specific to Fib-Tau.

p. 6 and Figure 2: The authors show in figure 2G that most tau aggregates co-localize with the plasma membrane after an exposure of 60 min. However, this might not apply for figure 2A, since more aggregates might be localized intracellularly with increasing exposure time. This aspect should be discussed in the legend or in the text. Ideally, the co-localization experiment should be done for different time points to assess uptake over time.

We assessed co-localization for longer time (4h) point to show that most of the “clusters” of Fib-Tau are on the plasma membrane (Appendix Figure 5, below **Fig D**). This does not mean that the fibrils are not endocytosed. Binding, clustering and endocytosis all occur at different rates. Because of their molecular weight, fibrils are expected to be endocytosed much slower than Tau monomers (Figure EV4).

Figure D. Majority of Fib-Tau clusters are at the cell surface (Appendix Figure S5)

p. 7 and figure 2H: The authors observe that only a few tau clusters remain on the cell membrane after washing and recovery of the cells for 60 min. They conclude that the tau fibrils either dissociate from the cells, or are taken up into the cells. However, they do not provide evidence for uptake versus dissociation to support this assumption. The fate of the fibrils after binding seems to be essential from a functional standpoint, and should be assessed by additional experiments, e.g. quantification of intracellular tau fibrils (labeled with a fluorescent dye).

We respectfully disagree with the reviewer statement “the fate of the fibrils after binding seems to be essential from a functional standpoint”. We do not assess endocytosis or the fate of Fib-Tau whether after internalization or dissociation. In this context, the statement we make to account for the fewer clusters we observe on cell membranes after washing the cells: “tau fibrils either dissociate from the cells, or are taken up into the cells” is open and well balanced.

p. 10 and figure 4F: It would make it easier for the reader if the authors could provide a figure that compares the major binding partners for 3R tau and 4R tau, and to what degree they overlap.

This is shown in Figure EV1C and Appendix Table S3 in the revised version of the manuscript.

p. 11: It would be helpful for the reader if the authors could include a schematic in the supplement, showing the polypeptide sequence of GluA2 and the localization of the crosslink with Fib-Tau. Also, were the authors able to derive any information on the tau residues that are involved in this crosslink? Please comment.

We appreciate this excellent suggestion. Two supplementary figures illustrate what the reviewer suggested (Appendix Figure S9 and S10).

Suppl. Figure 9: The increase of the calcium response upon exposure with tau is very subtle. What are the conclusions about the functional relevance of tau binding to the receptors, based on this experiment and the relatively limited impact on calcium signaling? The experiment should be repeated with higher tau concentrations, to see if this will increase the observed effects on calcium signaling.

This is Figure EV3 in the revised version of the manuscript. The reviewer correctly state that the effect on Ca^{2+} is weak. This is not surprising. Luckily, pathogenic assemblies do not affect target neurons immediately, otherwise the cells would rapidly die. This is definitely not what happens in our brains and tauopathies are age (i.e. time)-dependent progressive diseases. Higher concentrations of Fib-Tau will yield non-specific effects not necessarily related to their interaction with membrane proteins. To better assess to what extent Fib-Tau alter transmembrane ionic gradients, we performed Na^+ -imaging and assessed directly $\alpha 3$ -NKA activity (Figure EV5, corresponding text on Page 14). Neurons exposed to Fib-Tau have reduced capacity to pump Na^+ out of cells. This is direct evidence for impaired $\alpha 3$ -NKA activity.

Figure 7 and p. 12-14: It would increase the quality of the findings demonstrated with SPT-PALM if the authors could provide data for a control protein that does not interact with the receptors investigated here and thus does not induce the observed effects (e.g. reduction of the proportion of alpha3-NKA molecules in the Bound State upon exposure to Tau-Fib)

Additional measurements showing that the diffusion of $\alpha 3$ -NKA is not affected by monomeric Tau (Figure EV4, corresponding text on Page 13) have been performed. These data show that the effect we observe with fibrillar Tau is due to the fibrillar state of the protein.

p. 14, top of page, and figure 7J: The authors imply that Tau-Fib uptake might be mediated by clathrin-mediated endocytosis, since there is co-localization of Tau-Fib and clathrin pits in Figure 7J. It would enhance the impact of this observation if the authors could provide data for a functional relationship between NKA, clathrin, and Fib-Tau (e.g. by genetic manipulation of clathrin-mediated endocytosis, and/or the use of pharmacological inhibitors).

We appreciate the reviewer suggestion. We would like nonetheless to stress that reviewer is suggesting represent a separate full study and is beyond the scope of the present study.

p. 16, 2nd paragraph: The authors discuss that it might be "likely that the displaced NKA alpha3 subunit is endocytosed together with Fib-Tau clusters by neurons". The paper does not provide scientific evidence that this might be the case. Please change this statement accordingly or explain in more detail how this conclusion was made.

We provide evidence, through cell surface biotinylation experiments (Figure 7G-H), for a reduction in the amounts of $\alpha 3$ -NKA on neuron surfaces following exposure to Fib-Tau. We believe that this supports our claims.

Referee #3:

Accumulation of the tau protein is the defining pathological feature of a class of neurodegenerative diseases collectively referred to as tauopathies, that include Alzheimer's disease. A wealth of recent evidence suggests that tau oligomers can propagate in a prion-like manner. In this manuscript, Shrivastava et al. explore the binding of tau fibrils to neurons. Importantly, they found that tau fibrils interacted with the Na^+/K^+ -ATPase and AMPA receptors. They also show that tau clustering leads to a redistribution of the $\alpha 3$ subunit of Na^+/K^+ -ATPase and of the GluA2 subunit of the AMPA receptor. Finally, they show that α -

synuclein enhances tau clustering. While potentially interesting, the pathophysiological relevance of the data presented needs to be clarified.

Major points

- The Results section is not easy to read. It contains a large amount of practical information that should be restricted to the Materials and Methods section. The Results section should be extensively edited. The same comment also applies to some of the legends.

We have carefully read the manuscript and tried to simplify our sentences whenever possible. We did displace all practical information that does not impact understanding to the Material and Methods section.

- Figures 1C and 1D must be shown with the same image settings.

The images presented in panel 1C and 1D were independently acquired images under non-identical exposure setting at low magnification for qualitative display. This is stated in the legend. This is because, as shown in panel Figure 1E-F, the total fluorescence at 8h was much higher than 24h. The images in Figure 1E-F were acquired under identical exposure setting at high magnification.

- While investigating time-dependence of tau clustering makes sense (Fig. 2A), as it is a dynamic process, what is the pathological significance of investigating concentration-dependence? What would a physiologically meaningful concentration be? The same comment applies to α -synuclein in Figure 8.

The clustering of molecule interacting with neuronal membranes not only depends on time but also on their concentration. A high concentration of pathogenic protein on the plasma membrane will enhance clustering due to molecular crowding. The precise concentration of fibrillar phosphorylate or not Tau isoforms in the brain is in the range 50 to > 150 pg/ml depending on the study in various tauopathies e.g. 0.01 to 0.03pM. The concentrations we used throughout this study are in the nM range. Concentrations in the pM would be undetectable using the imaging techniques we used.

- It would be important to compare proteins binding to monomeric tau with those binding to fibrillar tau. Was this done?

We documented monomeric Tau binding and found that this form of the protein is rapidly internalized (within 1h). We present the data we obtained with monomeric Tau in the revised version of the manuscript (Figure EV4, and corresponding text on page 13 and 14). This includes α 3-NKA SPT-PALM measurements in the presence of monomeric Tau and the effect of Fib- α -Syn on monomeric Tau binding

- The immunoprecipitation validation of mass spectrometry data (Fig. 4E) is obviously essential. This must be done by Western blotting, a slot blot is not appropriate. Since detection of biotinylated fibrillar tau was done with streptavidin-HRP, how would IgGs interfere with detection? This should also be done for 1N3Rtau fibrils as well as 1N4Rtau fibrils.

We understand the reviewer's concerns. We explained in the figure legend, also the methods section (Appendix, Page 15), why we performed slot blots. IgGs heavy chains mask Tau immunodetection as they run at the same apparent molecular weight than Tau on SDS-PAGE. The reviewer can judge the problem we are facing from the blot below.

Figure E. Immunodetection of Tau by western blot

Attempt to immunodetect Tau using Tau 5 monoclonal antibody after immunoprecipitation of α 3-NKA and GluA2-AMPA receptors with an α 3-NKA goat and an anti GluA2 rabbit antibody upon exposure or not of neurons to Fib-Tau. The signal corresponding to the input Tau is shown on the right as is the signal coming from mouse IgG. Tau has an apparent molecular weight nearly identical to antibodies heavy chain.

We performed following the reviewer's suggestion co-immunoprecipitation experiments for Fib-Tau 1N4R exposed neurons (Figure EV1D). We observed that α 3-NKA and GluN1-NMDA receptors but not GluA2-AMPA receptors could be co-immunoprecipitated along with Fib-Tau-1N4R (Page 9).

- Why are data on 1N3Rtau fibrils presented in the main section of the manuscript and the data on 1N4Rtau as supplementary data rather than the other way round? This would be more logical as, although both isoforms are expressed in the adult human brain, 1N4Rtau is adult-specific.

The reviewer is correct, both isoforms are expressed in adult brain and both isoforms are found aggregated in Alzheimer's disease. The images we obtained with Tau 1N3R are of slightly better quality than those obtained with Tau 1N4R. This is why we present them in the main figures. The data obtained with Tau 1N4R are shown as Expanded View Figures. Overall very little differences are found between Tau 1N3R and 4R.

- The authors should discuss how heparin-induced tau filaments represent a good model for pathological tau species.

The reviewer raises an interesting question. Fibrils such as those used here are widely used. They are certainly way more relevant than bits and pieces of Tau harbouring mutations that are used in papers published in high impact journals. Besides being relevant in a review, but not here, discussing this issue is beyond the scope of this manuscript as we are not performing comparative studies for example with patients derived material which protein content, that may impact binding, is unknown.

Minor points

- The end of the Introduction is far too long. As it stands, it looks like an extended abstract of the whole manuscript. It should be limited to a few sentences stating the main results and their significance.

We have shortened the text from 2 pages to 1.5 page (e.g. by 25%).

- Legends of figures from Figure 2 should indicate that the experiments have been performed in cultured neurons.

We have added "primary" neurons in legends.

- 'string' should be written 'String' when referring to the database.
 Changed.

- Page 14. 'labs' should be 'laboratories'.
 Changed

2nd Editorial Decision

29th Oct 2018

Thank you for submitting your revised manuscript to The EMBO Journal. Your study has now been re-reviewed by the three referees and their comments are provided below.

As you can see the referees appreciate that the analysis has been strengthened, but they still raise some concerns that should be resolved in a final revision. Please make sure to improve the readability of the text and the discussion. The referees still find parts not clear enough and that some of the discussions point in the point-by-point response should be discussed in the text as well. Please also take a look at the points referee #3 brings up - they are important ones.

REFEREE REPORTS

Referee #1:

I have only a few minor points suggesting a thorough editing of the text:

(1) Both this reviewer and reviewer #3 noted that improvements to the text would increase the readability of this manuscript; however, very little changes have been done to the text (introduction and discussion as far as my assessment is concerned). Some of the discussion points in the rebuttal could have been incorporated in the main body of text as I assume the readers would have similar questions as I had. Similarly, the authors rarely specify in their point-by-point response where exactly they have made changes to the text and figures.

(2) Last sentence in the abstract needs to be changed: 'Our results demonstrate that fibrillar alpha-Synuclein and Tau cross-talk at the plasma membrane affects neuronal homeostasis and potentiate deleterious processes involved in Alzheimer and Parkinson's diseases onset.' What is the evidence that this cross-talk happens at the onset?

(3) Spelling: alpha-Synculein versus alpha-synuclein. Fig 1A,C,D typo: Pyramydal. Fig 1B Add '8h' for consistency. Fig 5D is not referenced in the text.

One additional point:

(4) Rebuttal point 1.7 is not clear: What do the authors mean by 'cellular processes' and not 'axons/dendrites'? Fig A: The image is taken above the plane of the coverslip. In the less denser areas there are lots of red dots outnumbering the yellow dots confirming what I argued about random precipitates (clusters) forming, some on the surface of the cell binding to axons and dendrites, and some floating around. In other words, it does not need the cellular membrane for these clusters to form, i.e. they form independently. I do appreciate that these precipitates form equally on axons and dendrites; whether they do bind to cell bodies is as far as I can tell not reported.

Referee #2:

This paper investigates the binding of exogenous 3R and 4R Tau assemblies (Fib-Tau) to the plasma membrane of mouse neurons in cell culture. The authors performed a proteomic screen using pull-down experiments to identify proteins of the neuronal plasma membrane that interact with exogenous applied Fib-Tau. They confirmed some of the hits with additional crosslinking/ LC-MS and co-immunoprecipitation and identified the alpha3 NKA subunit, the GluA2-AMPA and the

GluN1-NMDA as interactors with Fib-Tau. They subsequently demonstrate that Fib-Tau co-localizes with excitatory synapses and the identified cell membrane receptors. They employ live-neuron SPT-QD and SPT-PALM to show that the presence of Fib-Tau can affect the dynamics of the identified receptors at the plasma membrane. Finally, they show that the presence of alpha-synuclein fibrils enhances the clustering of Fib-Tau at the plasma membrane.

In the revised version of the manuscript, the authors have addressed all the major points raised in the previous review, and as a result the manuscript is substantially improved. More specifically, the authors have provided clarification on the general focus of the study, which is the analysis of tau fibril binding to the cell membrane and early functional events such as sodium and calcium flux across the membrane, but not the analysis of later events, such as the fate of the fibrils after binding (e.g. dissociation and/or uptake/endocytosis). Overall, the observations on co-localization and changes of membrane dynamics of several membrane proteins upon treatment with Tau-Fib made in this paper are novel and of interest for field and thus for the readership of EMBO. I still have some minor concerns that should be addressed before the publication of this manuscript, see below.

Specific comments:

p.4, last paragraph: The authors explain in the rebuttal that animals of the same gender were used for each timepoint of this experiment. I would suggest to present the data for both genders for each timepoint to avoid future questions regarding this aspect.

Figure 4 (and response to comment 12 by reviewer 1): Can the authors explain why only 29 out of 251 identified membrane proteins in the pull-down study have extracellular domains? Given that the study was designed to pull down proteins that interact with tau fibrils in the extracellular space, it is surprising that only ~10% of the identified proteins fulfil this criterium. Please clarify this aspect in the manuscript.

Response to my previous comment on the functional/pathophysiological relevance of this study (p.6 of the rebuttal) and discussion p. 17: As previously stated, I agree with the authors that this study is novel and describes interesting insights into the binding of tau fibrils to the membrane. However, by definition, the identification of a mechanism requires the description of a pathophysiological consequence (e.g. what is the impact on tau pathology). While the changes of calcium/sodium flux observed here are interesting, the study remains observational in this regard since it does not establish how relevant these changes are in the context of disease. Also, it does not seem realistic to design drugs that shield 29 binding partners on the cell membrane from tau fibril binding, as suggested in the discussion. Thus, from a pharmacological standpoint, further studies would be required to identify which ones (if any) of the binding partners are specific and functionally relevant for the interaction with tau and would be amenable to inhibition. For the publication of this manuscript, I suggest to address these aspects in more detail in the discussion.

Referee #3:

In their revised manuscript, Shrivastava et al. failed to answer satisfactorily some of the major points I made and their response actually raised more questions.

- I thank the authors for clarifying the concentration of fibrillar tau found in vivo. The very high, non-physiological concentrations of fibrillar tau used in their experiments are prone to cause non-specific effects. Reviewer #1 made an excellent point, what is referred to as clustering may well be random precipitation and sedimentation. The authors' response to this point is not convincing, and they should do what Reviewer #1 suggests, namely adding a control plate with no cells at all.
- If I understand correctly the blot included in the rebuttal, the secondary anti-mouse antibody (or the tau monoclonal?) cross-reacts with goat as well as rabbit antibody chains. This is seriously worrying and the authors should review their reagents and protocols. In any case, slot blots are not an acceptable way of validating interactions identified by mass spectrometry.

Point by point response to the Editor and reviewer's comments/requests

Thank you for submitting your revised manuscript to The EMBO Journal. Your study has now been re-reviewed by the three referees and their comments are provided below.

As you can see the referees appreciate that the analysis has been strengthened, but they still raise some concerns that should be resolved in a final revision. Please make sure to improve the readability of the text and the discussion. The referees still find parts not clear enough and that some of the discussions point in the point-by-point response should be discussed in the text as well. Please also take a look at the points referee #3 brings up - they are important ones.

We thank the Editor and Referees for their comments. We have carefully re-read the text and made additional changes with the aim of simplifying our sentences. All the changes we made are highlighted in blue. We had the feeling we did include the points that were interesting to include in the main text. We added additional issues we addressed in the first rebuttal, for example the point pertaining to the effect of crowding on the diffusion of Fib-Tau within the membrane. We also explain in the material section why we chose cortical neurons for biochemical experiments. Please let us know whether there is specific additional text you would like us to include.

Referee #1:

I have only a few minor points suggesting a thorough editing of the text:

(1) Both this reviewer and reviewer #3 noted that improvements to the text would increase the readability of this manuscript; however, very little change has been done to the text (introduction and discussion as far as my assessment is concerned). Some of the discussion points in the rebuttal could have been incorporated in the main body of text as I assume the readers would have similar questions as I had. Similarly, the authors rarely specify in their point-by-point response where exactly they have made changes to the text and figures.

We made changes to the text we felt adequate. We made additional changes. We shortened our sentences and made sure they are understandable. Most of the changes we made are highlighted in blue.

(2) Last sentence in the abstract needs to be changed: 'Our results demonstrate that fibrillar alpha-Synuclein and Tau cross-talk at the plasma membrane affects neuronal homeostasis and potentiates deleterious processes involved in Alzheimer and Parkinson's diseases onset.' What is the evidence that this cross-talk happens at the onset?

We thank the reviewer for bringing this issue up. The reviewer is correct. We bring no evidence for a potentiating effect. We have modified the last sentence of the abstract. As to the cross-talk between the two pathogenic assemblies, the finding that fibrillar alpha-synuclein favours the binding and clustering of Tau fibrils demonstrates cross-talk. We discuss how this may occur in the discussion section (page 18).

(3) Spelling: alpha-Synuclein versus alpha-synuclein. Fig 1A,C,D typo: Pyramidal. Fig 1B Add '8h' for consistency. Fig 5D is not referenced in the text.

" α -Synuclein" has been changed to " α -synuclein".

Fig 1. Pyramidal spelling has been corrected. In the same figure, panel 1G-H has been changed following Reviewer #2 comments.

One additional point:

(4) Rebuttal point 1.7 is not clear: What do the authors mean by 'cellular processes' and not 'axons/dendrites'? Fig A: The image is taken above the plane of the coverslip. In the less dense areas there are lots of red dots outnumbering the yellow dots confirming what I argued about random precipitates (clusters) forming, some on the surface of the cell binding to axons and dendrites, and some floating around. In other words, it does not need the cellular membrane for these clusters to form, i.e. they form independently. I do appreciate that these precipitates form equally on axons and dendrites; whether they do bind to cell bodies is as far as I can tell not reported.

Please refer to the response to referee 3 (point 1). We have replaced the term “processes” by “dendrites and axons” on page 6 to be precise.

Referee #2:

This paper investigates the binding of exogenous 3R and 4R Tau assemblies (Fib-Tau) to the plasma membrane of mouse neurons in cell culture. The authors performed a proteomic screen using pull-down experiments to identify proteins of the neuronal plasma membrane that interact with exogenous applied Fib-Tau. They confirmed some of the hits with additional crosslinking/ LC-MS and co-immunoprecipitation and identified the alpha3 NKA subunit, the GluA2-AMPA and the GluN1-NMDA as interactors with Fib-Tau. They subsequently demonstrate that Fib-Tau co-localizes with excitatory synapses and the identified cell membrane receptors. They employ live-neuron SPT-QD and SPT-PALM to show that the presence of Fib-Tau can affect the dynamics of the identified receptors at the plasma membrane. Finally, they show that the presence of alpha-synuclein fibrils enhances the clustering of Fib-Tau at the plasma membrane.

In the revised version of the manuscript, the authors have addressed all the major points raised in the previous review, and as a result the manuscript is substantially improved. More specifically, the authors have provided clarification on the general focus of the study, which is the analysis of tau fibril binding to the cell membrane and early functional events such as sodium and calcium flux across the membrane, but not the analysis of later events, such as the fate of the fibrils after binding (e.g. dissociation and/or uptake/endocytosis). Overall, the observations on co-localization and changes of membrane dynamics of several membrane proteins upon treatment with Tau-Fib made in this paper are novel and of interest for field and thus for the readership of EMBO. I still have some minor concerns that should be addressed before the publication of this manuscript, see below.

Specific comments:

1) p.4, last paragraph: The authors explain in the rebuttal that animals of the same gender were used for each time point of this experiment. I would suggest to present the data for both genders for each time point to avoid future questions regarding this aspect.

We now provide data from individual animals. The corresponding Figure 1G-H has been updated.

2) Figure 4 (and response to comment 12 by reviewer 1): Can the authors explain why only 29 out of 251 identified membrane proteins in the pull-down study have extracellular domains? Given that the study was designed to pull down proteins that interact with tau fibrils in the extracellular space, it is surprising that only ~10% of the identified proteins fulfil this criterium. Please clarify this aspect in the manuscript.

A membrane component includes all the proteins and protein complexes embedded/attached to the phospholipid layer. This includes scaffolding and signalling proteins. Pull down experiments identify not only the direct partners of Fib-Tau but also the protein complexes that interact with Fib-Tau binders. This is why we identify proteins without extracellular domain. Only proteins with extracellular domain are putative Fib-Tau-binding proteins. Figure 4 panel F illustrates what we describe. The proteins in yellow have extracellular domains. They interact in most cases physically with proteins in cyan.

Response to my previous comment on the functional/pathophysiological relevance of this study (p.6 of the rebuttal) and discussion p. 17: As previously stated, I agree with the authors that this study is novel and describes interesting insights into the binding of tau fibrils to the membrane. However, by definition, the identification of a mechanism requires the description of a pathophysiological consequence (e.g. what is the impact on tau pathology). While the changes of calcium/sodium flux observed here are interesting, the study remains observational in this regard since it does not establish how relevant these changes are in the context of disease. Also, it does not seem realistic to design drugs that shield 29 binding partners on the cell membrane from tau fibril binding, as suggested in the discussion. Thus, from a pharmacological standpoint, further studies would be required to identify which ones (if any) of the binding partners are specific and functionally relevant for the interaction with tau and would be amenable to inhibition. For the publication of this manuscript, I suggest to address these aspects in more detail in the discussion.

We thank the reviewer for this critical note. While we appreciate his/her pharmacological concerns and the importance of pathology, we also realize the fact that early events (minutes to hours) do not

need to explain pathology (a sum of events that occur much later on). We therefore inserted in the discussion section the sentence “Further studies may allow establishing a hierarchy amongst fibrillar Tau partners, thus defining protein interactions amenable to inhibition in view of therapeutic interventions.”. We do not wish to further develop this issue as we do not wish to over-interpret our findings.

Referee #3:

In their revised manuscript, Shrivastava et al. failed to answer satisfactorily some of the major points I made and their response actually raised more questions.

1) I thank the authors for clarifying the concentration of fibrillar tau found in vivo. The very high, non-physiological concentrations of fibrillar tau used in their experiments are prone to cause non-specific effects. Reviewer #1 made an excellent point, what is referred to as clustering may well be random precipitation and sedimentation. The authors' response to this point is not convincing, and they should do what Reviewer #1 suggests, namely adding a control plate with no cells at all.

We provide here examples of how Fib-Tau-1N3R-ATTO550 (0.36nM, 60min) distribution looks like on a coverslip, HEK cells (plated on coverslip) and hippocampal neuron (plated on coverslip and imaged within a region of low cell density). Few small spots are seen on the coverslip, but specific binding is detected when cells are present. We inserted this piece of data in Appendix Figure 4 following your request and that of reviewers #1 and 3.

- If I understand correctly the blot included in the rebuttal, the secondary anti-mouse antibody (or the tau monoclonal?) cross-reacts with goat as well as rabbit antibody chains. This is seriously worrying and the authors should review their reagents and protocols. In any case, slot blots are not an acceptable way of validating interactions identified by mass spectrometry.

We respectfully disagree with the reviewer. Slot blots are an acceptable way to validate interactions as a signal is detected when fibrils are present while no or a much weaker signal is detected in their absence. These data are in addition quantitative.

Nonetheless, we repeated the immunoprecipitation experiments and are providing in the revised version of the manuscript western blots (Figure 4E and EV1E) that further confirm the slot blot results in Figure 4E and Figure EV1D. The blots have been probed with streptavidin-HRP to circumvent the need for anti Tau and secondary antibodies that yield the results we presented in the first rebuttal.

3rd Editorial Decision

28th Nov 2018

Thanks for submitting your revised manuscript to The EMBO Journal. I have now had a chance to take a look at it and I appreciate the introduced changes. So I am happy with the paper!!

There are just a few things that we have to sort out before I can send you the formal acceptance letter.

Corresponding Author Name: Ronald MELKI

Manuscript Number: 99871